# Effect of Peierls-like distortions on transport in amorphous phase change devices
Nils Holle [1] ✉, Sebastian Walfort[1], Riccardo Mazzarello [2] & Martin Salinga [1] ✉

Today, devices based on phase change materials (PCMs) are expanding beyond their traditional application in non-volatile memory, emerging as promising components for future neuromorphic computing systems. Despite this maturity, the electronic transport in the amorphous phase is still not fully understood, which holds in particular for the resistance drift. This phenomenon has been linked to physical aging of the glassy state. PCM glasses seem to evolve towards structures with increasing Peierls-like distortions. Here, we provide direct evidence for a link between Peierls-like distortions and local current densities in nanoscale phase change devices. This supports the idea of the evolution of these distortions as a source of resistance drift. Using a combination of density functional theory and non-equilibrium Green's function calculations, we show that electronic transport proceeds by states close to the Fermi level that extend over less distorted atomic environments. We further show that nanoconfinement of a PCM leads to a wealth of phenomena in the atomic and electronic structure as well as electronic transport, which can only be understood when interfaces to confining materials are included in the simulation. Our results therefore highlight the importance and prospects of atomistic-level interface design for the advancement of nanoscaled phase change devices.

The growing amount of computing resources spent on training and inference of artificial intelligence systems jeopardizes corporate and, potentially, governmental goals for the reduction of carbon emissions. The need for increasing the efficiency of today's computing hardware at a pace that can keep up with this rapidly increasing demand is therefore evident. Computing in-memory promises to be a major leap towards more efficient computing hardware[1], as it circumvents the need to move data back and forth between memory and processing unit. With their high endurance, fast and reliable programming, and long retention times, memristive devices based on phase change materials (PCMs) offer very desirable properties to act as the basis of analogue devices in such in-memory computing hardware[2]. Fast switching between a highly conductive crystalline and less conductive amorphous phase facilitates the storage of information[3].

PCMs in their amorphous phase are subject to temporal evolution. As the molten material is rapidly quenched, it will eventually fall out of equilibrium, and form a glass. Like all other glassy systems, amorphous PCMs thus evolve towards the equilibrium supercooled-liquid phase. This relaxation process has been related to the resistance drift, where the already high resistance of the amorphous phase increases even further upon

physical aging[4–6]. In multi-level phase change storage, this limits the number of states that can be realized[7]. However, it can also be regarded as a "feature" rather than a "bug"[8], e.g. for the implementation of eligibility traces in neuromorphic circuits[9]. In either case, an understanding of and control over resistance drift is desirable in order to tune relaxation of the material as needed.

Several mechanisms have been proposed in the past to explain resistance drift, which includes disappearance of mid-gap states due to removal of homopolar Ge-Ge bonds[10,11], or a shift of the Fermi level towards the middle of the band gap[12]. We note that in principle, several different mechanisms could all contribute to resistance drift at the same time. However, one mechanism promises to be ubiquitous to all PCMs, which is the increase of Peierls-like distortions upon aging. Here, Peierls-like distortion refers to the formation of alternating short and long nearest neighbor bond distances around a central atom with approximately 180° bond angle[13,14]. These distortions characterize the crystalline phase, but have also been observed in the amorphous phase for many different PCM compositions, both in experiments and simulations[15–18]. Furthermore, there is evidence that the amount of distortions increases upon relaxation towards the supercooled liquid[10]. However, a direct link between an increase in Peierls-

[1]University of Münster, Institute of Materials Physics, Münster, Germany. [2]Sapienza Universitá di Roma, Department of Physics, Roma, Italy.
✉e-mail: nils.holle@uni-muenster.de; martin.salinga@uni-muenster.de

like distortions and an increase in device resistance has not been shown so far.

Here we show from first principles the influence of local variations in Peierls-like distortions on electronic transport. We observe higher current densities in regions with lower amounts of Peierls-like distortions. This gives direct evidence that an increase in Peierls-like distortions indeed increases device resistance. More distorted atomic environments have been observed upon structural relaxation of PCM glasses in a previous work[10], which links our results to resistance drift. For pure antimony, the suppression of resistance drift in 4 nm films has recently been connected to less distorted octahedral-like environments due to Sb/SiO$_2$ interfaces[19].

Simulations based on a combination of density functional theory and non-equilibrium Green's functions allow studying transport as a non-equilibrium phenomenon on an atomistic scale, without any empirical or semi-empirical model involved. We focus our study on pure antimony. Compared to more traditional PCMs like Ge$_2$Sb$_2$Te$_5$ or Ag$_4$In$_3$Sb$_{67}$Te$_{26}$, this material has several advantages, especially in aggressively miniaturized devices[20]. Using only a single element, there are neither demixing effects due to the strong electric fields present during switching, nor issues related to variations of stoichiometry and phase segregation[21] that limit cyclability in traditionally much more complex PCMs. Far more importantly for this study, however, the compositional simplicity allows focusing on effects of Peierls-like distortions alone, without distractions related e.g. to particular homo- or heteropolar bond motifs. The transferability of our results to more traditional PCMs will be discussed towards the end of our paper.

The idea that local structural variations in the amorphous PCM can affect electronic transport is not new. Models based on the assumption of randomly distributed energy barriers for Poole-Frenkel transport[22] have been explored in the past[23]. The resulting current density displayed percolative paths along the channel with the lowest activation energy. The conduction in amorphous PCMs has also been modeled with a Poole or Poole-Frenkel[24] approach and a multiple-trapping picture in other works[25–27]. These traps have been attributed to "wrong" homopolar bonds[10,28] or defective coordinations of germanium atoms[29] in compositions that include germanium. However, the very basic assumptions of Poole-Frenkel might not hold under all conditions in PCMs. Temperature and field dependent measurements imply intertrap distances in the range of 10 nm (GeTe, Ge$_2$Sb$_2$Te$_5$), or even 50 nm for Ag$_4$In$_3$Sb$_{67}$Te$_{26}$[26]. At latest in miniaturized devices smaller than 10 nm[30], we can hardly imagine a situation where this kind of model describes physical reality. In addition, with the advent of pure antimony as a phase change material[20,31–33], the above-mentioned pictures of electronic traps in the amorphous phase based on germanium atoms cannot hold for all PCMs. Understanding electronic transport in the next generation of nanoscaled devices thus calls for methods (and finally models) that are not based on electronic traps of questionable existence.

In this regard, most existing works on the transport in amorphous PCMs are either limited to ab-initio simulations of the ground-state electronic structure[10,29,34,35] or phenomenological models of transport based on experimental data of the temperature and field dependent conductivity[26,27,36]. While the former do not offer a direct connection from ground-state electronic properties to transport, the latter lack the link to the atomic structure offered by computer simulations based on density functional theory. So far, only very few works exist that study non-equilibrium transport in a phase change material atomistically. The oldest work is by Liu et al.[37]. The field-dependence of transport in both the metastable rocksalt and amorphous phase of Ge$_2$Sb$_2$Te$_5$ has been investigated by Roohforouz et al.[38]. Electromigration in GeTe and Sb$_2$Te$_3$ has been studied using non-equilibrium Green's functions (NEGF) simulations by Cobelli et al.[39]. However, the role of atomic structure in general and Peierls-like distortions in particular is not discussed in these works, and it is not obvious how results from GeTe, Ge$_2$Sb$_2$Te$_5$ or Sb$_2$Te$_3$ can be transferred to PCMs dominated by antimony. Note that in contrast to works that are restricted to the electronic structure of the bulk PCM, we also include effects of confinement through the metal electrodes. This has strong effects on both the atomic and electronic structure of the PCMs, which is because the presence of the electrodes

induces strong oscillations of mass density in the amorphous PCM. We will discuss this and other confinement effects further below.

## Results

### Heterogeneous electronic structure of amorphous antimony

To first elucidate the distribution of Peierls-like distortions in the super-cooled liquid and glass, we quenched from the melt a model of amorphous antimony with 728 atoms. We used a fixed volume that forces a density of 5.9 g cm$^{-3}$, as a previous work indicated that the disordered phase can be stabilized by decreasing mass density[20]. In addition, we have shown in another previous work on the atomic structure and electronic and optical properties of antimony in dependence of density[40] that the lowest density of 5.9 g cm$^{-3}$ studied there offers the largest amount of Peierls-like distortions, which promises to yield the largest effect on the electronic structure. The quenching rate of 9.5 K ps$^{-1}$ used here is the slowest rate that did not lead to an onset of crystallization during quenching in the study by Salinga et al.[20]. During quenching, we monitored the distances of the six nearest neighbors around each atom. These distances can be further grouped into three long and three short bonds. We also measured the magnitude of Peierls-like distortions using the angular-limited bond length ratio (details in the Methods section). This is the ratio of the average of the three longer distances and the average of the three shorter distances, limited to bonds that have an angle between 155° and 180°. We observe that Peierls-like distortions increase continuously in the supercooled-liquid and glass (Fig. 1b, d). As illustrated in Ref. 40, systems with mass densities up to the liquid density share the same continuous increase in Peierls-like distortions, albeit shifted to lower temperatures.

Distortions are not distributed homogeneously (Fig. 1c). Structurally correlated regions with different degrees of distortion evolve at low temperatures. Using a correlation function for the ratio $r_2/r_1$ of long to short bonds (details in the Methods section), we estimate that these regions extend up to approximately 1 nm at 500 K (Fig. 1e). Note that already at 500 K, the correlation function is most likely suppressed when exceeding length scales of approximately 7 Å due to finite size effects, as the size of the cubic simulation cell is approximately 2.9 nm.

At the fast quenching rate of 9.5 K ps$^{-1}$ used in our molecular dynamics simulation, the system must be expected to fall out of equilibrium at a rather high temperature. To make sure that the inhomogeneous distribution of distortions is not a consequence of this transition to a non-equilibrium state, we monitor potential energy as a function of temperature during melt-quenching, and as a function of time in additional constant volume (NVT) simulations at constant temperature (Fig. 2). Significant structural correlation in the distortion over approximately two atomic distances is already observed at 700 K (Fig. 1e). At this temperature, the system is still in equilibrium and does not show any sign of relaxation within 100 ps. At 500 K, the system is still rather close to the supercooled liquid and equilibrates within 100 ps. We therefore conclude that the structural correlation observed in Fig. 1e is a property of the supercooled liquid and not slowed by the glass transition.

We now focus on the consequences of the heterogeneous distribution of Peierls-like distortions on the electronic structure of supercooled-liquid and glassy antimony. Figure 3 shows the electronic inverse participation ratio (IPR) at the lowest simulated temperature of approximately 150 K. Again, the lowest temperature was chosen in order to more clearly see the effect of Peierls-like distortions on the electronic structure. In contrast to the structures at zero temperature that are usually studied, a temperature of 150 K is well in the range that has also been studied experimentally[20]. As shown in a previous work[40] using density-dependent ab-initio molecular dynamics simulations of antimony, the effect of further lowering the temperature is mainly a slight additional increase in Peierls-like distortions, as the thermal energy is not sufficient for larger reconfigurations. The inverse participation ratio measures the amount of atoms that are involved in a certain electronic state and, thus, the localization of the state itself. More precisely, the larger the IPR of a state, the smaller the number of atoms involved and the stronger the localization of the state. High values can be

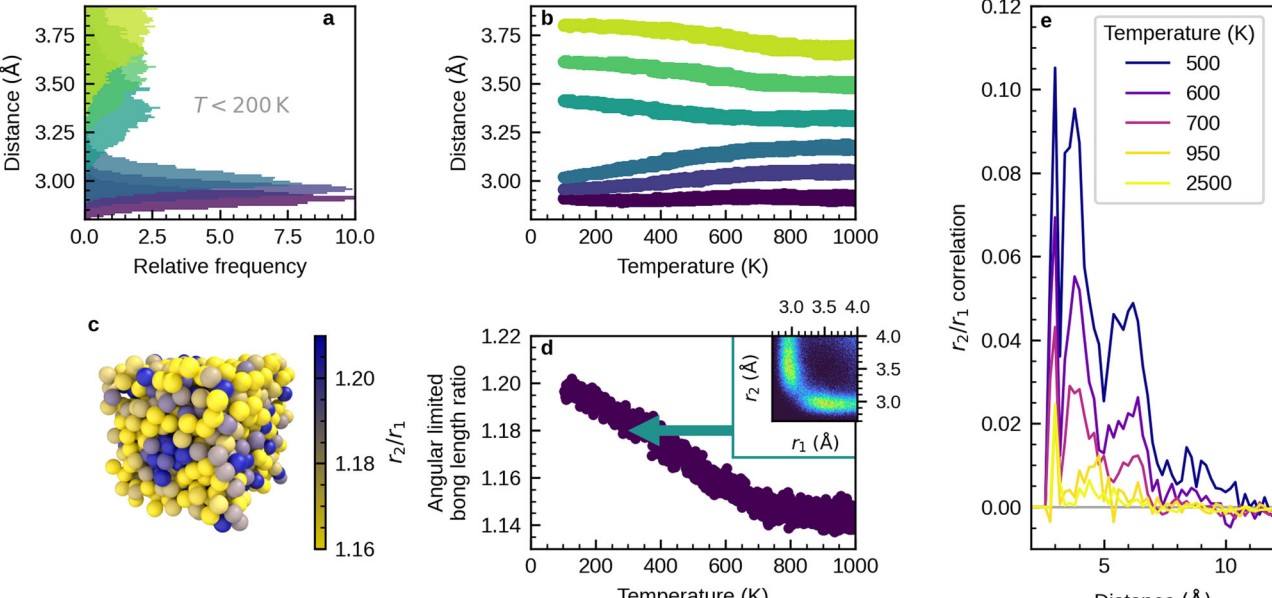

**Fig. 1 | Heterogeneous distribution of Peierls-like distortions in amorphous antimony. a** Distribution of distances of the six nearest neighbors at low temperatures. The atomic distances split into groups of three short ($r_1 \approx 3.0$ Å) and three longer ($r_2 \approx 3.2$ Å to approximately 4.0 Å) bonds. An exemplary illustration of Peierls-like distortions in the amorphous phase is shown in Supplementary Fig. 1. **b** Temperature dependence of the average distance of the six nearest neighbors. The clear splitting into long and short bonds emerges continuously below a temperature of 600 K. **c** Distribution of the ratio $r_2/r_1$ of long to short bonds at low temperatures shows correlated regions with low (light colors) and high (dark colors) amounts of distortion. **d** The angular-limited bond length ratio again illustrates the continuous increase of Peierls-like distortions in the supercooled liquid and glass. The angular-limited bond length ratio is defined as given in the Methods section. The inset shows the similar and more common angular-limited three body correlation at room temperature, again as defined in the Methods section. **e** The structural correlation length of distortion increases beyond the nearest neighbor distance at temperatures below 700 K. The correlation function is defined in the Methods section. The correlation length is likely influenced by finite size effects already at 500 K, as the size of the simulation box is only 2.9 nm.

observed around "band tails" at the Fermi level and at $E = E_F \pm 4.5$ eV, which indicates localized states. The insets visualize the spatial distribution of selected states in the vicinity of the Fermi level. While we see high amplitudes of the wavefunction in regions with low amounts of Peierls-like distortions (light colors in Fig. 1c), atoms in more distorted environments (dark colors in Fig. 1c) do not take part in these states. This already indicates a link between the distribution of Peierls-like distortions and electronic structure.

For a more quantitative assessment, we visualize the temperature dependence of the average magnitude of distortion in each electronic state (Fig. 3b). Details of the calculations are again given in the Methods section. At energies where we observed localization of electronic states in the inverse participation ratio, we now find lower amounts of Peierls-like distortions than the average. This can be seen particularly well for the lowest temperature of 150 K, and around the Fermi level. The same observation holds for the "band edges" in general, albeit with no consequences for transport. States around the Fermi level are thus localized in less Peierls-like distorted regions. Note that this holds for all states between $-0.5$ eV and 0.5 eV around the Fermi level, with localization increasing continuously towards $E_F$. At the same time, the average ratio of $r_2/r_1$ decreases continuously, with the lowest value also found at $E_F$. Furthermore, we find very few states that are especially localized directly at the Fermi level. As indicated by the visualization of the wavefunction of these states (Fig. 3a), these states are also found in regions with lower amounts of Peierls-like distortions. This must be expected as Peierls-like distortion open the pseudo-gap and lead to reduction of the number of states at the Fermi level, which means that the remaining states can be found in less distorted regions.

**Influence of Peierls-like distortions on the local current density**
So far, we have only discussed the ground-state electronic structure of bulk antimony in the disordered state. Electronic transport, however, is a complex non-equilibrium phenomenon that cannot be fully understood based

on ground-state quantities. As we now aim to find a direct connection between atomic structure and electronic transport properties, the method used must also describe transport as the non-equilibrium phenomenon that it is. To this end, we melt-quench another model of antimony comprising approximately 1100 atoms that includes tungsten electrodes. Due to its high melting point and structural stability compared to other electrode materials, tungsten is a very attractive choice as an electrode material for nanoscaled phase change devices with high endurance and has also been used in commercial phase change memory[41] as well as experimental devices[20]. The two outermost tungsten layers were fixed in the molecular dynamics simulations in order to resemble the bulk structure of tungsten. Our device setup allows studying transport through liquid, supercooled-liquid and finally glassy antimony directly using the non-equilibrium Green's function (NEGF) method, without resorting to the (semi-)empirical models for the connection between electronic structure and transport discussed in the introduction section. An exemplary snapshot from our simulations as well as the heterogeneous distribution of Peierls-like distortions in the device setup is shown in Fig. 4a.

We used an NEGF/density functional theory method (details in the Methods section) to calculate the current density through our phase change device structure. Note that as transport is described in a steady-state approximation, we cannot include effects like trapping and emission of charges from defect centers. Instead, defects act as scattering centers here. From the self-consistent solution to the non-equilibrium scattering problem, we calculate the local current density. We find regions of high (red) and low (white) current density in the glassy phase change material (Fig. 4b).

Furthermore, we observe clear effects of confinement by the tungsten electrode in our amorphous device structure. The local density of states (Fig. 4c) exhibits a layered structure. Note that this effect is not due to (partial) crystallization, but a property of the confined antimony liquid, supercooled liquid and glass. It can be traced back to confinement-induced

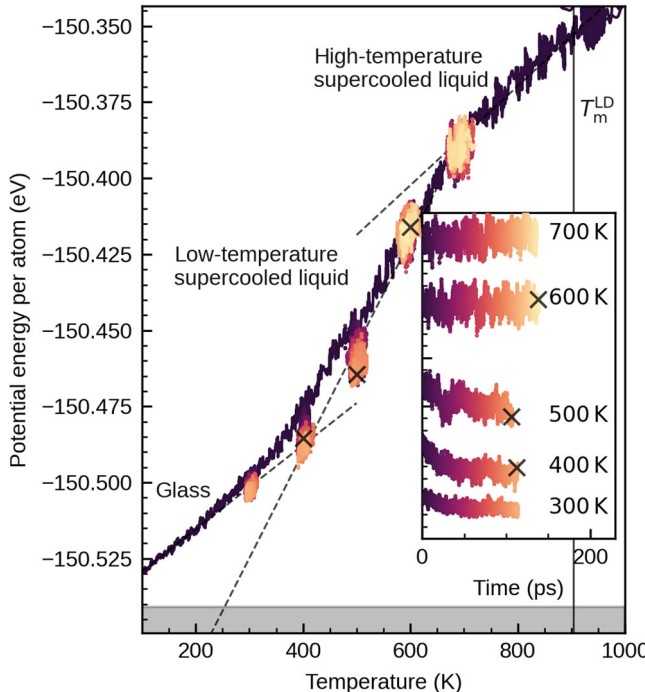

**Fig. 2 | Heterogeneous distribution of Peierls-like distortions is an equilibrium property and is observed in the supercooled-liquid.** The potential energy for the model with a mass density of 5.9 g cm$^{-3}$ is shown in dependence of temperature during melt-quenching in ab-initio MD simulations with a quenching rate of 9.5 K ps$^{-1}$ (dark line). At several different temperatures, branches with constant temperature are created. Here, color indicates the time starting at departure from the melt-quenching trajectory. While full equilibration is observed directly after quenching at 600 K and 700 K, the system begins to deviate from the supercooled-liquid line at approximately 500 K. Note that the quenching rate used here is approximately 14 orders of magnitude larger than the rate typically used to determine a glass transition temperature in DSC measurements. This is why the system falls out of equilibrium at a higher temperature. $T_m^{LD}$ marks the (experimental) melting point of antimony at a density of 6.49 g cm$^{-3}$. The dashed lines indicate the approximated dependence of potential energy on temperature of the supercooled liquid in equilibrium at high and low temperatures and a linear extrapolation of the low-temperature behavior of the glassy phase. An approximation of the supercooled liquid line at lower temperatures is obtained by a linear fit of the potential energies at 400 K to 600 K after equilibration. Note that the system is probably not fully equilibrated after 100 ps at 400 K and this datapoint should be at slightly lower energies. For visual clarity, the data in the main area were smoothed using a Savitzky–Golay filter. The raw data for constant temperatures between 300 K and 700 K are depicted in the inset. The gray area indicates an upper bound estimate of the energy of the crystal, which is obtained from the recrystallized sample at 400 K.

oscillations of mass density (Supplementary Fig. 3), which are also observed e.g. in colloidal systems[42]. These oscillations decay towards the center of the device, and are not simply due to stress-induced effects (see Supplementary Fig. 4-7). In addition, we observe that liquid antimony forms a strongly bound wetting layer on the tungsten (100) surface already at very high temperatures, which is thus also present in the supercooled liquid and glass. Mass density oscillates around the average density of the full volume of 6.33 g cm$^{-3}$, which also includes the wetting layer. Without the wetting layer, we find a slightly smaller value of 6.25 g cm$^{-3}$, which is also approximately the density in the very center of the device (see Supplementary Fig. 3). The wetting layer follows the BCC structure of the tungsten electrode (Supplementary Fig. 8), which is in line with the experimental results from Ref. 43. The wetting layer is then followed by a rather large gap to the remaining, disordered antimony. Note that this "gap" is not due to the amorphous material "shrinking" and thus assuming a rather high density at the center, but also exists at the interface of crystalline antimony and tungsten (100) (see Supplementary Fig. 9). In the Supplementary Information, we show that this

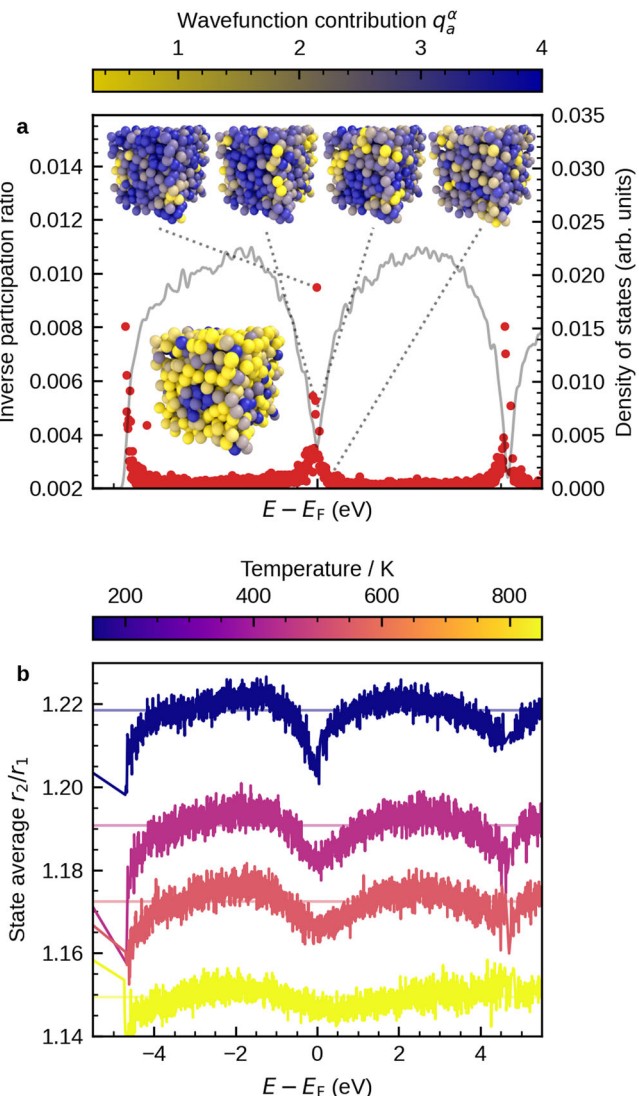

**Fig. 3 | Electronic states at the Fermi level are localized in regions with lower amounts of Peierls-like distortions. a** Density of states (gray line) and inverse participation ratio of each state (red dots) at a temperature of approximately 150 K. The insets at the top visualize the spatial distribution of selected electronic states around the Fermi level. They are ordered by their value of the inverse participation ratio (IPR) from high values (left) to low values (right). Light and dark colors indicate high and low wavefunction amplitudes of a specific state, respectively. We observe localization in regions with low amounts of distortions. Compared to the inset at the bottom left, which shows again the distribution of the ratio $r_2/r_1$ of long to short bonds at low temperatures (light colors) and high (dark colors) amounts of distortion (Fig. 1c). In the dark, distorted region at the front center, we observe no wavefunction amplitude for any of the localized states shown at the top, but some amplitude for the delocalized state (top right). The rightmost inset at the top shows a delocalized state for comparison, where the distribution of amplitudes is homogeneous. Isosurface plots of the same states are shown in Supplementary Fig. 15. **b** Ratio of long to short bonds ($r_2/r_1$) averaged for each electronic state in dependence of energy (details in the Methods section). The results are shown for four different temperatures. Horizontal lines indicate the average value of $r_2/r_1$ at the respective temperature. The localized states observed in (**a**) show lower amounts of Peierls-like distortions than the average. This is true in particular for states around the Fermi level. As also described in the Methods section, additional reference calculations of the density of states (DOS) and IPR using the more accurate TB09 exchange-correlation functional are shown in Supplementary Fig. 2.

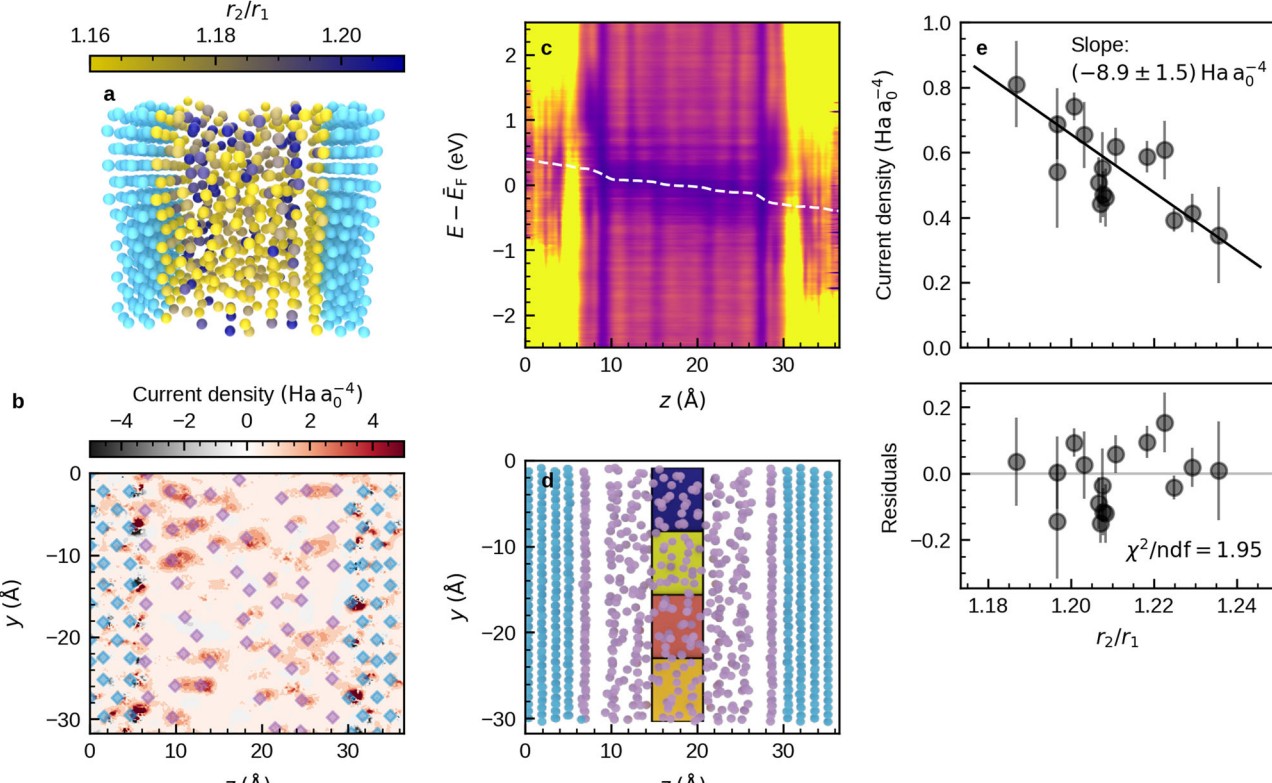

**Fig. 4 | Influence of Peierls-like distortions on local currents. a** Peierls-like distortion are heterogeneously distributed in the phase change device. Light colors depict atoms in less distorted environments (low $r_2/r_1$). **b** Current density in $z$-direction shows more and less conductive paths in the PCM. The current density was calculated for a single molecular dynamics snapshot at 250 K with a bias of 0.8 V, and is visualized for a slice through the device at $x = 10$. Red colors indicate current in the direction of transport, while black colors indicate reverse current due to backscattering. The diamonds indicate atomic positions. **c** Local density of states (LDOS) shows formation of layers in amorphous antimony due to confinement by the tungsten electrodes. Again, the LDOS at a bias of 0.8 V is calculated for a single snapshot at approximately 250 K from the ab-initio molecular dynamics simulation. The white line indicates the local electrostatic difference potential $\delta V_E$, where the electrostatic difference potential at zero bias was subtracted. The values at $z = 0$ and

$z = l$ (with $l$ the device length) correspond to the chemical potential of the respective electrode. Note that this potential is not completely flat in the contacts, probably due to the limited number of electrode layers and the limited number of $k$-points, which are both imposed by the computational demand of our simulations. **d** Antimony atoms in the center of the device are split into $4 \times 4$ groups. Average $r_2/r_1$ and average local current are calculated for each group as described in the main text. Only the first row of the $4 \times 4$ grid of groups is visible. Color indicates the average local current density in each group, where light and dark colors indicate high and low current density, respectively. **e** A large current density correlates with low values of distortion ($r_2/r_1$). The average current density in each group of atoms illustrated in (**d**) is calculated from ten different snapshots at 250 K, each 100 fs apart. Again, a bias of 0.8 V is used. The result is plotted versus the average distortion $r_2/r_1$ in each group. Error bars indicate the standard deviation.

larger distance is due to a very different bonding situation compared to bulk antimony (Supplementary Fig. 9). Bonding from the antimony wetting layer to the tungsten surface shows a more ionic character due to the difference in electronegativity between tungsten and antimony. The bonds to the next layer of antimony are rather weak. As transport across the gap could now be dominated by small local fluctuations in gap size rather than effects of distortion, we take from the local density of states that we should restrict the following analysis to only the very center part of the device. This is because we want to focus on the effects of Peierls-like distortion alone, which might be hidden if the local current is dominated by a few bonds to the wetting layer with high conductivity.

To now elucidate the connection between atomic structure and current density, we partition antimony atoms in the very center of the device into 16 different subgroups (see Fig. 4d). For each group, we calculate the average current density and average amount of Peierls-like distortions (details in the caption of Fig. 4). Plotting current density vs. distortion (Fig. 4e), we observe a clear correlation between both quantities. Less distorted regions in our device structures carry relatively more current, which is in line with the larger number of states at the Fermi level in these regions. A linear fit to the data yields a slope that deviates from zero by five standard deviations, thus establishing that the local current density does in fact depend on Peierls-like distortions.

## Discussion

A physical picture that arises from the combination of our findings is the following: The conduction in the amorphous phase of antimony is subject to local variations due to a heterogeneous distribution of Peierls-like distortion. Regions with low amounts of distortion have a relatively high density of states (DOS) around the Fermi level, a link that is illustrated in Ref. 40. The limited size of these regions thus imposes a limit on the extent of states around the Fermi level, which renders these states localized. There is evidence that as PCM glasses relax towards the supercooled liquid, the magnitude of Peierls-like distortions increases, which is connected to an increase of the size of the band gap[10]. With the present work, we now provide evidence from first-principles calculations of electron transport that this increase does in fact contribute to resistance drift. If the magnitude of distortion increases with time, this also leads to a lower amount of regions with higher conductivity (Fig. 4e), and overall increase in device resistance. The factor of two that we observe between local current densities in the regions with highest and lowest amounts of distortion (Fig. 4e) is of the order of magnitude that one would usually expect for resistance drift in PCMs like Ge2Sb2Te5[44,45] on the timescale of tens of seconds.

Note again that other mechanisms could also contribute to this increase in resistance. Besides effects related to "wrong" homopolar bonds[10,28] or defective coordinations of germanium atoms[29], this could in

principle also include effects due to the presence of crystalline inclusions in the amorphous material, which have been proposed to be related to a possible conduction mechanism in the past[25]. However, even in the presence of crystalline regions, the total device resistance through a series of amorphous and crystalline regions should still be heavily dominated by the amorphous part. This is because of the orders of magnitude higher resistivity of these regions. Any changes of the conduction within the purely amorphous regions should thus be much more important for resistance drift than any effect related to crystalline inclusions.

In this regard, we can also transfer our results to other, more traditional PCMs like $Ge_2Sb_2Te_5$, GeTe or $Ag_4In_3Sb_{67}Te_{26}$. As long as Peierls-like distortions are present in the amorphous PCM, very similar effects on electronic transport should be observed. However, other structural mechanisms that depend more on the exact elements involved could also contribute to resistance drift in these materials. As a reminder, mechanisms proposed in the past include e.g. the disappearance of mid-gap states due to removal of homopolar Ge-Ge bonds[10,11], yet not all PCMs contain germanium, or a shift of the Fermi level towards the middle of the band gap[12]. Pure amorphous antimony does not show a band gap at 250 K in our simulations. This is in contrast in particular to $Ge_2Sb_2Te_5$ and GeTe, where a significant gap is observed at 0 K and mid-gap states are often the focus of studies of the electronic structure[10,29,34,35]. The low density of our antimony model increases the width of the pseudo-gap[40] and should thus move our results closer to those for antimony-rich compositions like $Ge_{15}Sb_{85}$ or $Ag_4In_3Sb_{67}Te_{26}$. Furthermore, a reduction of the band gap size due to temperature is often ignored in electronic structure calculations of other PCMs, such that an actual gap might not even exist over the full range of operating temperatures of PCMs[46]. We believe that simulations of electronic transport in other PCMs at elevated temperatures are needed to disentangle the contributions of the described variety of mechanisms to resistance drift. Devices based on pure antimony are the ideal study object for a first step towards this disentanglement: An overall increase in Peierls-like distortions localizes states around the Fermi level, and finally decreases local currents. The device resistance should thus increase as Peierls-like distortions increase with time.

Although we believe that this finding is an important step towards a complete understanding of transport in amorphous antimony and other PCMs, and the NEGF method used here goes significantly beyond studies based on ground-state electronic properties, many important effects are still neglected. This also reflects in a large discrepancy between the resistance of our simulated device at 250 K and the resistance of experimental devices[20]. The data in Ref. 20 suggest a specific resistance of the amorphous thinfilm with a thickness of 5 nm in the range of $10^{-3}$ Ωm (assuming that the full melting zone was amorphized, which is not necessarily the case). We find a value of the order of $10^{-5}$ Ωm in our simulations (see Supplementary Fig. 11-12). For the crystalline phase, we find a conductivity of $1.2 \times 10^6$ S m$^{-1}$ (Supplementary Fig. 12-13), which is much closer to experiments ($2.5 \times 10^6$ S m$^{-1}$). Note that a comparison between the theoretical and experimental results for amorphous antimony must be done with care, as the interface situation was different in the experiments in Ref. 20. In contrast to our simulations, there was no direct interface between amorphous antimony and tungsten in the experiments, but only an interface between amorphous antimony and crystalline antimony. Furthermore, the Perdew-Burke-Ernzerhof exchange-correlation functional used in our density functional theory calculations overestimates the number of states at the Fermi level and underestimates their localization, which leads to larger values of the total current. This holds in particular for the amorphous structures at low temperatures.

Still, probably the most important effect neglected here is the scattering of electrons on vibrational modes. Simulations of GeTe that include this scattering indicated only a small influence of phonon scattering on the current-voltage curve[47]. However, these results were obtained assuming scattering on only a single phonon mode with a single electron-phonon coupling strength, which cannot reflect scattering that includes the full vibrational mode spectrum. As changing bond lengths due to local Peierls-like distortions will affect the local vibrational spectrum, in particular through shorter and thus more rigid short bond lengths, there should also be consequences for the local conductivity. With different local vibrational spectra, local scattering rates will also differ.

Despite all these limitations, we can already learn a lot from the NEGF calculations shown here. This is not limited to the influence of Peierls-like distortions on electronic transport. With Fig. 4, we underline in multiple ways that to understand transport in nanoscaled phase change devices, simulations of the bulk PCM alone are by far not sufficient. First of all, we show that confining antimony to a thin film with a thickness of only a few nanometers does not only increase the stability of the amorphous phase[20], but also leads to spatial oscillations of mass density in the supercooled-liquid and glassy state, which affects the electronic structure (Fig. 4c). Furthermore, the formation of a wetting layer on the tungsten surface and subsequent gap to the actual amorphous material should depend strongly on the electrode material. Significant voltage drops across this gap between the wetting layer and amorphous antimony (see white line in Fig. 4c). The size of this gap will therefore have a large influence on the temperature and bias dependence of current through the device. However, it does not impede our finding of a connection between Peierls-like distortions and local current, as we limited our study to the central region of the device. Both the density oscillations and influence of the wetting layer cannot be studied by simulations of the bulk amorphous phase[48], or confinement by dense layers of antimony[49] or the crystalline PCM[50]. Our results indicate that future studies should take much more care of both effects, as they promise a rich set of additional phenomena in miniaturized phase change devices with respect to electronic, optical, and dynamical properties.

Supplementary Fig. 14 shows that despite the strong oscillations of mass density, the influence of Peierls-like distortions on local current density can also be observed closer to the interface, but possibly with a less steep dependence that depends on the local contacts to the wetting layer. This demonstrates that very close to the electrode interface, the (volume) mass density might not be a good measure to predict atomic structure and behavior of the PCM. This is in contrast to the amorphous phase, where Peierls-like distortions can be tuned by changing mass density[40] or pressure[18]. In bulk amorphous antimony, mass densities of more than 6.8 g cm$^{-3}$ already lead to a complete removal of Peierls-like distortions[40]. Here, densities of more than 10 g cm$^{-3}$ are observed locally, with Peierls-like distortions still visible. Our simulations indicate an interesting formation of a stack of amorphous 2D structures of antimony above the wetting layer. These structures could also show optical and dynamical properties that are very different from those of both the bulk amorphous phase, and crystalline antimonene.

As already indicated when discussing the heterogeneous distribution of Peierls-like distortions in the beginning, the cell size is limited to approximately 3 nm in all directions. While this extent is already rather large for ab-initio molecular dynamics simulations, it is still small compared to phase change cells studied in experiments with sizes often exceeding tens of nanometers. In contrast to the situation in radically miniaturized devices described in the introduction, where conduction based on electronic traps is questionable, localized states can well determine transport characteristics in larger structures. Furthermore, we have to expect more localized states around the Fermi level for PCMs with an actual band gap like $Ge_2Sb_2Te_5$ or GeTe. Thus, it could happen in large enough systems that the system size exceeds the localization length due to localization of states in less Peierls-like distorted regions. The formation of regions with lower amounts of Peierls-like distortion could then be a generic mechanism for electron localization in PCMs, which is not related to particular bond motifs[10,11] or defective coordinations[29] of germanium. These localized states could in turn act as traps for Poole-Frenkel conduction that has been proposed as the conduction mechanism in $Ge_2Sb_2Te_5$ or GeTe[25–27]. Studies on much larger systems and other PCMs will be needed to clarify this.

Coming back to our original goal of relating resistance drift to Peierls-like distortions, we note that resistance drift is a statistical process. Besides the relevance of this phenomenon for applications, noise in phase change

materials is also interesting as resistance fluctuations can be linked to the underlying potential energy landscape of the material[51]. The relation of structural features to resistance fluctuations, however, is not well understood. From the connection of Peierls-like distortions to local current densities shown above, it is evident that Peierls-like distortions could also play a role for resistance fluctuations in the glassy state. Compared to the study presented here, many more simulations of the total and local currents would be needed to simulate noise on very short timescales from first principles. Note that even if this is accomplished, comparison to experiments will be limited due to the very limited time scales of ab-initio simulations of a nanosecond or less. Advanced methods like machine-learned potentials will thus be needed to study noise on time scales that are comparable to experiments. Even then, this will be a very challenging task, but could ultimately allow connecting resistance fluctuations observed in experiments to changes in the atomic structure.

We conclude that atomistic simulations of electronic transport in nanoscaled PCMs hold great promise to unravel the connection between atomic structure of PCMs on the one hand, and measurements of more macroscopic electronic transport characteristics in experiments on the other. We showed that rather subtle changes in local atomic structure in the form of Peierls-like distortions can change the local current by a factor of two (Fig. 4e), which is of the order of magnitude that is usually observed for resistance drift on time scales of tens or hundreds of seconds in traditional PCMs like $Ge_2Sb_2Te_5$[44,45] at room temperature. This gives direct evidence that an increase in Peierls-like distortions indeed increases device resistance. An increase in distortions has been observed upon structural relaxation of PCM glasses in a previous work[10], which links our results to resistance drift. For pure antimony, the suppression of resistance drift in 4 nm films has recently been connected to less distorted octahedral-like environments due to $Sb/SiO_2$ interfaces[19]. We can therefore link changes in Peierls-like distortions to resistance drift. The factor of two change in local currents mentioned above indicates a sizable contribution of this effect to the overall drift, but further investigation is needed in particular for multicomponent PCMs like GeTe or $Ge_2Sb_2Te_5$, where different effects have also been proposed.

Our study also highlights the important role of the effects of nanoconfinement and interfaces in phase change devices scaled to very small dimensions of only a few nanometers. With very little phase change volume, not only the confinement itself, but also the exact orientation and condition of interfaces from the PCM to surrounding materials will become important. The accurate and reliable fabrication of such nanoscaled devices, however, needs fabrication techniques that give control over the phase change volume on the level of only a few thousand atoms or fewer. In view of all the prospects of advances in phase change technology through nanoscaling that we mentioned in the beginning, we believe that this should be the focus of future research in this area, both in simulations and experiments. Achieving this level of precision will be a key enabler for next-generation phase change devices, and bring atomistic simulations and experimental studies of electronic transport in the same structures within reach.

## Methods
### Ab-initio molecular dynamics simulations
We used the 2nd generation Car-Parrinello method as implemented in the Quickstep code[52] of CP2k[53] for all our ab-initio molecular dynamics simulations[53]. The scheme combines the efficiency of Car-Parrinello simulations with the large time steps used in Born-Oppenheimer molecular dynamics. A Langevin thermostat was employed to control temperature. The time step was set to 2 fs. For simulations of pure antimony, the number of atoms in the cubic simulation cell was kept constant at 728. Finite size effects can significantly influence dynamics in the system and have been discussed in a previous work[20]. Simulations with tungsten electrodes included a total of 1104 atoms (528 antimony, 576 tungsten). The positions of atoms in the two outermost tungsten layers were kept fixed during the simulation in order to be able to attach crystalline, semi-infinite electrodes for the transmission studies. We checked that this does not affect the

temperature distribution in the remainder of the device (Supplementary Fig. 10). As it was overall difficult to reach the thermostat temperature with the 2nd generation Car-Parrinello method in our interfacial system, we increased the number of corrector steps in the always stable predictor-corrector scheme[54] to eight. The density functional theory calculations were carried out using the PBE functional[55], a wave function cutoff of 300 Ry, Goedecker-Teter-Hutter pseudopotentials[56], and triple-/double-$\zeta$-valence-polarized basis sets for antimony and tungsten, respectively.

### Definition of the angular-limited bond length ratio
The *angular-limited three-body correlation* (ALTBC) is a higher-order correlation function that includes three atoms. It is defined as[57]

$$g^{(3)}(r, r') = \frac{V}{\rho(N-1)(N-2)} \times \sum_{i_1} \sum_{i_2} \sum_{i_3} \langle \delta(r - r_{12}) \delta(r' - r_{23}) \Theta(\beta - \delta) \rangle,$$

with $i_1$, $i_2$ and $i_3$ distinct indices of atoms, $V$ the unit cell volume, $N$ the number of particles, $\rho = V/N$ the atomic density,

$$\cos \beta = \frac{\vec{r}_{i_1 i_2} \cdot \vec{r}_{i_2 i_3}}{r_{12} r_{23}}$$

the alignment angle and $\delta$ an angular threshold (typically 25° in phase-change materials). Based on this correlation function, we define the *angular limited bond length ratio* as

$$\text{ALBLR} = \frac{\sum_{i_1, i_2, i_3} \frac{\max(r_{12}, r_{23})}{\min(r_{12}, r_{23})} \Theta(\beta - \delta) \bar{\Theta}(r_{12}) \bar{\Theta}(r_{23})}{\sum_{i_1, i_2, i_3} \Theta(\beta - \delta) \bar{\Theta}(r_{12}) \bar{\Theta}(r_{23})},$$

with $\bar{\Theta}(r) = \Theta(r - r_{\min}) \Theta(r_{\max} - r)$. The cutoff radii were set to $r_{\min} = 2.5$ Å and $r_{\max} = 4$ Å, respectively.

### $r_2/r_1$ correlation function
We calculated a spatial correlation function of $r_2/r_1$ as

$$q_{\text{Peierls}}(r) = \left\langle \langle s(0)s(r) \rangle - \langle s(0) \rangle \langle s(r) \rangle \right\rangle_0,$$

where

$$s(r) = \begin{cases} 1, & \text{for } (r_2/r_1)(r) \geq (r_2/r_1)_{\text{thresh}} \\ -1, & \text{for } (r_2/r_1)(r) < (r_2/r_1)_{\text{thresh}} \end{cases}.$$

Here, $(r_2/r_1)(r)$ denotes the ratio of long to short bond distance of an atom at distance $r$. $r_1$ is the average distance of the three nearest neighbors, $r_2$ the distance of the three next-nearest neighbors. We used $(r_2/r_1)_{\text{thresh}} = 1.185$ as a threshold value to distinguish between atoms in distorted and undistorted environments.

### Electronic structure calculations
The electronic structure of amorphous antimony was calculated using a linear combination of atomic orbitals (LCAO) density functional theory method implemented in QuantumATK[58]. The density of states and inverse participation ratio were calculated on a $3 \times 3 \times 3$ Monkhorst-Pack grid of $k$-points using the exchange-correlation functional by Perdew, Burke, and Ernzerhof[55] (PBE), a Fritz Haber Institute (FHI) pseudopotential and corresponding basis set of double-$\zeta$-double-polarized (DZDP) quality. The energy cutoff was set to 150 Ha. Note that semi-local exchange-correlation functionals like PBE tend to underestimate band gaps. However, the reduction in state-average Peierls-like distortions in Fig. 3 is observed over a very large range of temperatures in the supercooled-liquid, long before an actual band gap is observed. Our results thus do not depend on an accurate estimation of the band gap. Nevertheless, we performed additional reference

calculations using the more accurate exchange-correlation function of Tran and Blaha (TB09)[59]. The results are shown in Supplementary Fig. 2. We observe qualitatively the same behavior as for the corresponding PBE calculations shown in Fig. 3a. At the lowest temperature of 150 K, we see more states with exceptionally strong localization than the PBE results, which had to be expected, but not larger maximum values of the inverse participation ratio (IPR).

State-average quantities can be calculated as follows: The IPR for a non-orthogonal basis set and a system of $N$ atoms can be defined as[60]

$$\mathrm{IPR}_\alpha = \sum_{a=1}^{N} |q_a^\alpha|^2,$$

with the contribution to state $\alpha$ from atomic site $a$ given by

$$q_a^\alpha = \sum_{b_1, a_2, b_2} \mathrm{Re}\left[ S_{ab_1 a_2 b_2} c_{ab_1}^{\alpha*} c_{a_2 b_2}^\alpha \right].$$

Here, $S$ is the overlap matrix $S_{ij} = \langle \phi_i | \phi_j \rangle$, and $c$ denotes the wave function coefficients $\psi_\alpha(\vec{r}) = \sum_i c_i^\alpha \phi_i(\vec{r})$. This contribution can also be visualized directly, which can be seen in the insets in Fig. 3.

Suppose that we now have some structural quantity $X_a$ that we can define for each atom. In analogy to the IPR, the corresponding "state-average" quantity $X_\alpha$ is then defined as

$$X_\alpha = \sum_a X_a q_a^\alpha.$$

In this case, we use

$$X_a = \frac{r_2^a}{r_1^a},$$

where $r_1^a$ is the average distance of neighbors 1-3, and $r_2^a$ the average distance of neighbors 4-6, sorted by distance.

## Non-equilibrium Green's functions calculations

Based on single snapshots from our ab-initio MD simulations of antimony/tungsten structures, we performed non-equilibrium Green's functions calculations of electron transmission and current. We again used the LCAO density functional theory method in QuantumATK for this purpose. Semi-infinite tungsten electrodes were attached on both sides of the simulation cell, with a BCC lattice parameter of $a = 3.165$ Å. The transmission spectrum was then calculated in dependence of bias, sampling the electronic structure at the $\Gamma$-point only. We again employed an FHI pseudopotential, using a basis set of DZP and DZDP quality for tungsten and antimony, respectively. The energy cutoff was set to 110 Ha. The self energy was calculated using a recursion method that exploits inherent sparsity[61]. The current was finally calculated from the transmission function $T(E)$ using

$$I(\mu_L, \mu_R, T_L, T_R) = 2\frac{e}{h} \int T(E) \left[ f\left( \frac{E - \mu_R}{k_B T_R} \right) - f\left( \frac{E - \mu_L}{k_B T_L} \right) \right] dE,$$

where $\mu_{L/R}$ and $T_{L/R}$ are the left and right chemical potential and electrode temperatures, respectively, and $f$ is the Fermi-Dirac function. The electrode temperatures are chosen to match the snapshot temperature.

Besides the transmission function, a number of other quantities can be extracted from the non-equilibrium density matrix. The current density was obtained using

$$\mathbf{J}(r) = -\frac{e\hbar}{4\pi m} \int dE \sum_{\mu,\nu} D_{\mu,\nu}(E) \phi_\nu(r) \nabla\phi_\mu(r) \left( f\left( \frac{E - \mu_L}{k_B T_L} \right) - f\left( \frac{E - \mu_R}{k_B T_R} \right) \right),$$

where $D_{\mu,\nu}$ is the spectral density matrix, and $\phi$ denotes the atomic basis orbitals[62].

## Data availability
The data that support the findings of this study are available from the corresponding author upon reasonable request.

## Code availability
The code used to calculate the angular-limited bond length ratio is publicly available at https://zivgitlab.uni-muenster.de/ag-salinga/fastatomstruct.

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

## Acknowledgements

We acknowledge support from the German Research Foundation (DFG) through the collaborative research centers Nanoswitches (SFB 917) and Intelligent Matter (SFB 1459) as well as the European Research Council (ERC-Grant 640003). The calculations were performed using resources provided by the CIT of the University of Münster (Palma II HPC cluster).

## Author contributions

N.H. performed the computer simulations and analyzed the data. N.H., S.W., R.M., and M.S. contributed to the interpretation and compilation of the results in a series of discussions in the course of this study. N.H. wrote the manuscript with input from S.W., R.M., and M.S.

## Funding

## Competing interests

The authors declare no competing interests.
