## [Transparent Peer Review file · Communications Materials]

Effect of Peierls-like distortions on transport in amorphous phase change devices

Corresponding Author: Professor Martin Salinga

Version 0:

Decision Letter:

Dear Professor Salinga,

Thank you for submitting your manuscript, "Effect of Peierls-like distortions on transport in amorphous phase change devices", to Communications Materials. It has now been seen by 2 referees, whose comments are appended below. You will see that while they find your work of potential interest, they have raised substantial concerns that must be addressed. In light of these comments, we cannot accept the manuscript for publication, but are interested in considering a revised version that addresses these serious concerns.

We hope you will find the referees' comments useful as you decide how to proceed. Should further experimental data or analysis allow you to address these criticisms, we would be happy to look at a substantially revised manuscript. However, please bear in mind that we will be reluctant to approach the referees again in the absence of major revisions. If the revision process takes significantly longer than three months, we will be happy to reconsider your paper at a later date, as long as nothing similar has been accepted for publication at Communications Materials or published elsewhere in the meantime.

When submitting your revised manuscript, please include the following:

-A response letter with a point-by-point reply to each of the referee comments and a description of changes made. Please include the complete referee report in the response letter. Please note that the response letter must be separate to the cover letter to the editors.

-A marked-up version of the manuscript with all changes to the text in a different colored font. Please do not include tracked changes or comments. Please select the file type 'Revised Manuscript - Marked Up' when uploading the manuscript file to our online system.

-A clean version of the manuscript. Please select the file type 'Article File'.

-An updated <https://www.nature.com/documents/nr-editorial-policy-checklist.zip> Editorial Policy checklist, uploaded as a 'Related Manuscript File' type. This checklist is to ensure your paper complies with all relevant editorial policies. If needed, please revise your manuscript in response to these points. Please note that this form is a dynamic 'smart pdf' and must therefore be downloaded and completed in Adobe Reader. Clicking this link will download a zip file containing the pdf.

Please use the following link to submit your revised manuscript files:

Link Redacted

Please do not hesitate to contact me if you have any questions or would like to discuss the required revisions further. Thank you for the opportunity to review your work.

Best regards,

Reinhard Maurer, PhD
Editorial Board Member
Communications Materials
orcid.org/0000-0002-3004-785X

Reviewers' comments:

Reviewer #1 (Remarks to the Author):

In this paper, the authors show that a resistance drift in the amorphous phase (towards higher resistance) can result from Peierls distortion. With a focus on single-element Antimony-based PCM, they establish such a link with the following observations from computational results: (1) Peierls distortion (split into regular long- and short- bonds in the lattice) occurs during the quenching process, and (2) there is a correlation between lower Peierls distortion and higher current density. The slow increase in distortion in the amorphous material over time can thus slowly shut down these current pathways, thus leading to an increase in resistance over time.

The results presented seem interesting, but the manuscript could use be further improved:

General points:

 The discussion focuses on the purely amorphous phase of the material. However, resistance drift in PCM is mainly an issue in Multi-Level-Cells (MLCs) which try to achieve several intermediate states. Such intermediate states may have crystalline and amorphous domains, and the current would probably be carried dominantly by electronic states corresponding to the crystalline regions. If this is the case, is the contribution of atomic distortion processes within the purely amorphous regions still a very relevant effect for the resistance drift observed in practice? The authors specifically compare their findings to the magnitude of resistance drift observed in PCMs (page 10, paragraph 2), so a discussion on this would be relevant.

 The writing is very wordy. For example, too many sentences are used to justify an ab initio approach to transport. The explanation of temperature effects on the bandgap can also be shortened.

Comments:

 The authors state that "results obtained using GeTe might not be transferable to PCMs dominated by antimony", but also that "This gives direct evidence that the increase in Peierls-like distortions upon structural relaxation of the PCM glass shown in [10] is indeed a source of resistance drift.", where Ref [10] uses GeTe. Maybe it would be better to cite experimental studies which show some evidence of Peierls distortion in Antimony. For example, the following paper: <https://onlinelibrary.wiley.com/doi/full/10.1002/adv.202301043>

 Figure 1 needs some significant improvements. The figures and font are all too small, part (b) needs a y-axis label (it seems to be the distance, but an axis should still have a label). There must be a way to show the information in part (c) in a clearer way.

 The distortions exist in clusters, rather than being homogeneously distributed, but the authors state that it might be influenced by the small cell size. How large is this influence? Can these clusters be a complete artefact of the finite simulation box?

 Could the mass density oscillation effect be due to internal pressure during the MD simulations, from fixing the positions of the outer contact atoms (which implicitly imposes a fixed cell)? Does it remain if a full geometry-relaxation is performed?

 It seems that the potential is not flat in the contacts (Fig. 4c). Maybe this is due to computational limitations on the number of contact atom layers which could be included, but it should be addressed in the text.

 "Peierls-like distortions can change the local current by a factor of two (Fig. 4 e), which is of the order of magnitude that is usually observed for resistance drift ... This gives direct evidence that changes in Peierls-like distortions upon structural relaxation of the glass give a relevant contribution to resistance drift. " - I think this is quite a large jump. See the 'general comments' section above.

 In Fig. 4b, the current is negative in the electrodes and positive in the PCM. How can that result in a netto positive or negative current? Or is that an artefact of the plotting scheme?

Minor comments:

 use `` instead of " for the left open bracket in LaTeX

 paragraph 3: one mechanisms  one mechanism

 There are several similar minor typos

 "Structurally correlated regions with smaller and larger amounts of distortions evolve at low temperatures." - maybe the authors can rephrase this

 Some of the details behind how distortion was measured/defined should be shifted into the main manuscript, since this is important to follow the text

 "We find more and less conductive paths through the glassy phase change material (Fig. 4 b). " - It is not clear what this means, even after looking at Fig. 4b. The authors may mean to say something like "regions of high (red) and low (blue) current density".

 Some important information seems to be located in a manuscript the authors currently have under review elsewhere. Maybe phrases like "As shown in Ref [37]" should be rewritten to "as shown using ... [37]" so the results can be more easily understood in isolation

 "we melt-quench another model of antimony comprising approximately 1100 atoms that includes tungsten electrodes" - It should be clarified here that the outermost tungsten positions were fixed.

 "We use a DFT-based method (details in the Methods section) to calculate the current density " - maybe this should be rephrased, it sounds like DFT is solving the non-equilibrium problem.

 "but also exists at the interface of crystalline antimony and tungsten (100). " - Citation missing (this should probably be [3] from the Supplementary Materials)

Reviewer #2 (Remarks to the Author):

In this manuscript, Holle et al. performed ab initio simulations of non-equilibrium transport in a model of amorphous antimony (Sb), by combining density functional theory (DFT) with non-equilibrium Green's function (NEGF) calculations. It is widely accepted that electronic transport in the amorphous state of phase-change memory materials is not very well understood, and in-depth investigations, at the atomic level, of all the relevant aspects that govern the transport in such memory devices are missing from the literature. In addition, transport can be correlated to the time-dependent resistance drift phenomenon observed in the amorphous phase of these materials.

Here, the authors are trying to draw connections between atomistic models of amorphous phase-change materials and macroscopic electronic transport characteristics observed in the devices. They show that small changes in the local atomic structure, through Peierls-like distortions, can change rather significantly the local current density, which, in turn, can be correlated with the resistance drift typically observed in the device. Hence, they conclude that such Peierls-like distortions should play a role into the structural relaxation of the amorphous state (Sb in this case as the amorphous structure). In addition, the authors showcase the importance of studying the effects of nano-confinement in phase-change memory devices, and how this can affect all the properties of the material, while also they discuss the relevance of interfaces inside the devices which are very small in size.

The research is very interesting, while the authors have extensive experience in studying phase-change memory materials with experimental and modelling techniques, providing useful contributions to the relevant community in understanding the processes inside phase-change memory devices. The authors provide technical details about the atomistic simulations carried out here, the electronic structure properties of amorphous Sb, the NEGF calculations, and the results about the electronic transport. Also, I appreciate the significant effort in the Discussion section of the manuscript to provide a comprehensive picture about the limitations of this study, the comparisons with different compositions (binary, ternary, etc.), the connections between the results and experimentally-based observations, as well as the suggestions for future work related to the concepts presented here.

Below, I express some comments, concerns and suggestions that I would like to ask the authors to consider:

i. The authors mentioned in the last paragraph of the Introduction that "the only existing work that studies non-equilibrium transport in a phase-change material is by Liu & Anantram [36]." a statement related to previous atomistic simulations available in the literature. Apologies if I am mistaken, but I think this is not probably 100% true, since this is not the only existing study. Previously, electronic transport in amorphous and crystalline Ge₂Sb₂Te₅ has been studied with ab initio molecular-dynamics simulations and NEGF calculations [AIP Advances 9, 055120 (2019)]. In addition, ab initio simulations have been combined with NEGF calculations to study electromigration processes in liquid GeTe and Sb₂Te₃ [J. Phys.

Chem. C 124, 9599-9603 (2020)]. Again, these studies seem relevant to me, in case I am not wrong.

ii. The authors highlight in the Introduction the necessity for “models and methods that are not based on electronic traps of questionable existence”, while they also say that in miniaturized devices the concept of trapping cannot really describe physical reality. I can imagine that the miniaturization from a device engineering point of view might exclude such processes. But then, the authors in the results section of the manuscript discuss “localized states” in their amorphous Sb model. If here one refers to the spatial electron localization of an electronic state (such as those that appear in a glass near the bottom of the conduction band or the top of the valence band), then a “localized state” can potentially serve as a charge-trapping centre inside the glass structure. I think it would be interesting for the reviewers and the (future) readers if the authors could kindly clarify this.

iii. In Figs. 1(a) and 1(b) the authors show the atomic distances of three short and three longer bonds. For the longer bonds they state a range of values, but the shorter bonds they seem to be almost identical. My question is if this is meaningful. Why do you need to define “three different” short bonds that are almost the same? In other words, what is the statistical significance of these three different short bonds?

iv. In Fig. 1(e), the specific choice of temperature values of 942 K and 2502 K is rather striking. It might sound quite particular, but why not 940 and 2500 K?

v. In page 5, the authors bring into the surface the issue of “finite size effects” regarding the extension of the distortions within the periodic cubic simulation cell. They have modelled a glass structure of 728 atoms, which is already quite large for ab initio simulations. Could the authors provide an estimation or a view of what size of glass model should be adequate for such quantifications? Size effects is usually an “easy argument” (still valid), but if one puts it into the context of what is the size needed for the investigations then it becomes more meaningful, especially for comparisons. In other words, a discussion between a qualitative and quantitative comparison.

vi. The choice of the quench rate for the generation of the amorphous Sb model seems a bit arbitrary (9.5 K/ps). Could the authors provide a brief justification regarding this?

vii. The choice of colour for the text and dash lines in Fig.2 is completely inadequate. It is very difficult to read that text and distinguish the lines in the figure with using this silver-like colour. I strongly recommend to change this, since the readers cannot see those (especially the text written).

viii. In the analysis presented in Fig. 2, the authors extracted specific snapshots of the MD trajectory at temperatures during the quench of the simulated structure, and then run longer MD simulations to equilibrate the model at these temperatures. I am wondering if the authors considered to perform NPT (constant temperature, constant pressure) MD simulations at this stage. The quench simulation is usually adding a residual stress to the simulation box. If one performs an NPT simulation initially for the extracted geometry at the specific temperature, it should help the structure to adjust at the “simulation volume” that is happy to find, and then it could equilibrate smoother. In that way, the structure should find its way to the respective equilibrium volume as well. It would be interesting to hear what is the view of the authors, and how this could affect their structures (especially the Peierls-like distortions) and results? To my mind, this should be relevant regarding such distortions.

ix. The authors throughout the manuscript they make a rather big statement/argument that in contrast to traditional DFT calculations performed at 0K, they have performed simulations at temperatures that are well in range with the experimentally (device operation level) temperatures. I can see the point and I am happy to follow it, especially for higher temperatures, close to the liquid or super-cooled liquid regimes. But then, the authors use this argument for their analysis of a structure at 150 K. I would be a little bit skeptical to believe that a model structure of a glass at 150 K would look significantly different (geometrically) from the same structure quenched to 0 K, for example. MD simulations at these temperatures typically they do not yield at different atomic structures (essentially atoms do not have much chance to move). Unless for amorphous Sb the differences in the atomic structure for temperatures between 0 – 300 K are that significant. I think the authors need to consolidate this argument with the atomistic models that have generated.

x. The same concept also applies for the band gap of amorphous Sb. My first question is: does one expect for amorphous Sb to have a band gap (such as GeTe, for example)? Of course, I follow the argument of the authors that in higher temperatures the band gap of the model is “shrinking”, but in lower temperatures (such as 300 K or 150 K) will the system naturally have a band gap? Because if the answer is yes, then the methodology of choice to study the electronic structure and relevant properties is relatively inadequate for making strong arguments. The authors have discussed this already in the manuscript (at least for the IPR calculations) with respect to the DFT GGA approach, while also they performed calculations with a more accurate, meta-GGA functional. This is all good. But, could it be possible a calculation with a hybrid functional (i.e., inclusion of the Hartree-Fock exchange) to give a result of a model of amorphous Sb with having a well-defined band gap?

xi. The way that the authors decided to plot the wavefunctions of selected electronic states in the inset of Fig. 3(a) is rather vague. In the caption it is mentioned that “light and dark colours indicate high and low amplitudes, respectively”. But, what is exactly high/low in this case? The authors need to provide a range or values. Moreover, wavefunctions (atomic orbitals) of electronic states obtained from DFT calculations are typically drawn as iso-surfaces, a concept that is useful in order to discuss electron localization or delocalized states. To me, it seems that these figures have been generated with a continuous colour palette within a range of values, which makes it difficult to really assess localization within the amorphous network. Also, the choice of blue and the colouring scheme makes everything difficult to compare. The changes among the

snapshots seem very fractional (difficult to see), which then makes difficult to grasp how these connections and arguments could hold true. In other words, the character of localized states the authors claim in their analysis seems very weak, i.e., are the electrons spatially localized in a region inside the glass?

xii. There is a typo in page 8. The text reads: "The local density of states (Fig. 4) exhibits a layered structure". I think it should be Fig. 4c.

xiii. Throughout the manuscript the authors chose to show their atomistic models with colouring the atoms using a specific palette to describe properties. It would be interesting to see a picture of the atomic configuration as well of the generated models. How do the Sb local environments look like in their model? How do these Peierls-distortions look like at the atomic level?

xiv. My final question is about the transferability regarding the validity of the results by using different DFT packages. The authors presented in the manuscript and supplementary material calculations performed with three different codes (CP2K, Quantum ATK, Quantum Espresso). All these codes can do DFT calculations. CP2K utilizes a mixed Gaussian and plane-waves method with pseudopotentials. Quantum Espresso is strictly plane waves with pseudopotentials. Quantum ATK combines pseudopotentials with LCAO and plane-wave basis sets. A question arises how the different calculations with the different codes compare to each other? The pseudopotentials at each code should be different, while the philosophy of the DFT calculations is also different. Please do not get me wrong, I do not think that the calculations are problematic, but one needs to be aware of this, especially from the point of view that these are numerical calculations.

The research presented in this manuscript is certainly very topical. The current study contributes to the efforts to rationalize the electronic transport and resistance drift in amorphous phase-change materials. In addition, the authors provide useful insights for future research directions in the field, which is very important for continuation. At the same time, within the current manuscript, it is necessary for some things to be clarified, and provide a more coherent picture.

Overall, I would be happy to potentially recommend the publication of this manuscript in Communications Materials, after the authors have tried to respond to my comments and suggestions.

Communications Materials is committed to improving transparency in authorship. As part of our efforts in this direction, we are now requesting that all authors identified as 'corresponding author' create and link their Open Researcher and Contributor Identifier (ORCID) with their account on the Manuscript Tracking System prior to acceptance. ORCID helps the scientific community achieve unambiguous attribution of all scholarly contributions. You can create and link your ORCID from the home page of the Manuscript Tracking System by clicking on 'Modify my Springer Nature account' and following the instructions in the link below. Please also inform all co-authors that they can add their ORCIDs to their accounts and that they must do so prior to acceptance.

Version 1:

Decision Letter:

Dear Professor Salinga,

Thank you for submitting your revised manuscript, "Effect of Peierls-like distortions on transport in amorphous phase change devices", to Communications Materials. It has now been seen again by the 2 referees, whose comments are appended below. You will see that while they find your revised work interesting, Reviewer #2 has some remaining concerns that should be addressed before we can offer to publish your study. We therefore invite you to revise and resubmit your manuscript, taking into account the points raised via a suitable rebuttal letter, and incorporating further comments in the revised version to clarify the unanswered questions.

Please also note that Figure 2 appears to be essentially identical to Fig. 3 of the recently accepted PRL paper [40]. Please

note that you will likely need to request the relevant copyright from APS should you wish to leave the figure as is.

When submitting your revised manuscript, please include the following:

-A response letter with a point-by-point reply to each of the referee comments and a description of changes made. Please include the complete referee report in the response letter. Please note that the response letter must be separate to the cover letter to the editors.

-A marked-up version of the manuscript with all changes to the text in a different colored font. Please do not include tracked changes or comments. Please select the file type 'Revised Manuscript - Marked Up' when uploading the manuscript file to our online system.

-A clean version of the manuscript. Please select the file type 'Article File'.

-An updated <https://www.nature.com/documents/nr-editorial-policy-checklist.zip> Editorial Policy checklist, uploaded as a 'Related Manuscript File' type. This checklist is to ensure your paper complies with all relevant editorial policies. If needed, please revise your manuscript in response to these points. Please note that this form is a dynamic 'smart pdf' and must therefore be downloaded and completed in Adobe Reader. Clicking this link will download a zip file containing the pdf.

In the event that your manuscript is accepted we will provide detailed guidance on our journal policies and formatting. You may however wish to ensure that the manuscript complies with our house style at this stage. See our style and formatting guide (<https://www.nature.com/documents/commsj-phys-style-formatting-guide-accept.pdf>) and checklist (<https://www.nature.com/documents/commsj-phys-style-formatting-checklist-article.pdf>) for reference.

Data availability statements and data citations policy: All Communications Materials manuscripts must include a section titled "Data Availability" at the end of the Methods section or main text (if no Methods). More information on this policy, and a list of examples, is available at <http://www.nature.com/authors/policies/data/data-availability-statements-data-citations.pdf>.

- Accession codes for deposited data
- Other unique identifiers (such as DOIs and hyperlinks for any other datasets)
- At a minimum, a statement confirming that all relevant data are available from the authors
- If applicable, a statement regarding data available with restrictions
- If a dataset has a Digital Object Identifier (DOI) as its unique identifier, we strongly encourage including this in the Reference list and citing the dataset in the Data Availability Statement.

DATA SOURCES: We strongly encourage authors to deposit all new data associated with the paper in a persistent repository where they can be freely and enduringly accessed. We recommend submitting the data to discipline-specific, community-recognized repositories, where possible and a list of recommended repositories is provided at <http://www.nature.com/sdata/policies/repositories>.

If a community resource is unavailable, data can be submitted to generalist repositories such as <https://figshare.com/> or <http://datadryad.org/> Dryad Digital Repository. Please provide a unique identifier for the data (for example a DOI or a permanent URL) in the data availability statement, if possible. If the repository does not provide identifiers, we encourage authors to supply the search terms that will return the data. For data that have been obtained from publically available sources, please provide a URL and the specific data product name in the data availability statement. Data with a DOI should be further cited in the methods reference section.

Please use the following link to submit your documents:

Link Redacted

We hope to receive your revised paper within three months; please let us know if you aren't able to submit it within this time so that we can discuss how best to proceed. If we don't hear from you, and the revision process takes significantly longer, we will close your file. In this event, we will still be happy to reconsider your paper at a later date, as long as nothing similar has been accepted for publication at Communications Materials or published elsewhere in the meantime.

Please do not hesitate to contact me if you have any questions or would like to discuss these revisions further. We look

forward to seeing the revised manuscript and thank you for the opportunity to review your work.

Best regards,

Reinhard Maurer, PhD
Editorial Board Member
Communications Materials
orcid.org/0000-0002-3004-785X

Reviewers' comments:

Reviewer #1 (Remarks to the Author):

The authors satisfactorily addressed my comments. I am happy with the revised version of the manuscript.

Reviewer #2 (Remarks to the Author):

In the rebuttal document, the authors considered carefully all the requests and concerns raised by both the reviewers, providing very detailed answers and explanations. Then, a portion of these answers was inserted in the revised manuscript. A major part of the answers was based on evidence and analyses from another accepted manuscript (in Phys. Rev. Lett.) and a second (yet) unpublished study. I think this is okay, but it is interesting to highlight that for some of the arguments constructed in this manuscript there was a need for extra studies.

Overall, this is a very interesting paper and it corresponds to another piece of the puzzle that the authors (and others) are trying to solve regarding the resistance-drift issue in amorphous phase-change memory devices. Therefore, I would be happy to recommend the publication of the revised manuscript in Communications Materials.

Nevertheless, for the sake of the scientific discussion I have some further comments and observations regarding the responses:

Reviewer 2 Q10: The authors provided further evidence and calculations of higher level of theory to show that their model of amorphous Sb does not have a band gap (essentially a metallic behaviour). I agree and this is all fine, as well as very nicely demonstrated. But one of the questions in the Reviewer report still remains to be answered: "Does one expect for amorphous Sb to have a band gap?", in other words what is the experimentally measured value (if any) of the band gap for the material under study?

Reviewer 2 Q11: The authors argued against plotting the wavefunctions obtained from the DFT calculations by utilizing isosurfaces using two reasons in their response. Argument (a) is valid and I get it - the fact that they are trying to compare them consistently with the visualizations of the Peirls-like distortions. However, with argument (b) I beg to differ - if one uses isosurfaces to claim spatial localization for electronic states that are delocalized, then there is something fundamentally wrong in the logic of the user. And as the authors argued in their favour later on, the IPR cannot lie.

Reviewer 2 Q13: If the authors would like to follow a "statistical" approach, then simulations in many independent models are required to gather statistics. For amorphous materials it is challenging to capture the probability of bond formation and to obtain distributions of their properties via a single numerical study. But, I am sure the authors are in the right path regarding this with their follow-up ML-based study.

Reviewer 2 Q14: The mention of the LOBSTER code seems new to me here. I might have missed it somewhere with respect to the analysis presented in the manuscript.

Communications Materials is committed to improving transparency in authorship. As part of our efforts in this direction, we are now requesting that all authors identified as 'corresponding author' create and link their Open Researcher and Contributor Identifier (ORCID) with their account on the Manuscript Tracking System prior to acceptance. ORCID helps the scientific community achieve unambiguous attribution of all scholarly contributions. You can create and link your ORCID from the home page of the Manuscript Tracking System by clicking on 'Modify my Springer Nature account' and following the instructions in the link below. Please also inform all co-authors that they can add their ORCIDs to their accounts and that

they must do so prior to acceptance.

Version 2:

Decision Letter:

Dear Professor Salinga,

Your manuscript titled "Effect of Peierls-like distortions on transport in amorphous phase change devices" has now been seen again by our editorial team. In light of our assessment of the latest revisions and rebuttal, I am delighted to say that we are happy, in principle, to publish a suitably revised version in Communications Materials.

We therefore invite you to revise your paper one last time to address the remaining concerns of our reviewers. At the same time we ask that you edit your manuscript to comply with our journal policies and formatting style in order to maximise the accessibility and therefore the impact of your work.

EDITORIAL REQUESTS

* Your manuscript should comply with our policies and format requirements, detailed in our style and formatting guide (<https://www.nature.com/documents/commsj-phys-style-formatting-guide-accept.pdf>).

* Please edit your manuscript according to the editorial requests in the attached table, and outline revisions made in the right hand column. If you have any questions or concerns about any of our requests, please do not hesitate to contact me. It is important that each request be addressed in order to avoid delays in accepting your manuscript. Please upload the completed table with your manuscript files as a Related Manuscript file.

* The editorial requests table also includes a full list of the files that must be provided upon resubmission. Please upload your files according to this table.

* An updated editorial policy checklist that verifies compliance with all required editorial policies must be completed and uploaded with the revised manuscript. All points on the policy checklist must be addressed; if needed, please revise your manuscript in response to these points. Please note that this form is a dynamic 'smart pdf' and must therefore be downloaded and completed in Adobe Reader. Clicking this link will download a zip file containing the pdf.

OPEN ACCESS

Communications Materials is a fully open access journal. Articles are made freely accessible on publication. For further information about article processing charges, open access funding, and advice and support from Nature Research, please visit <https://www.nature.com/commsmat/open-access>

Please use the following link to submit your revised files:

Link Redacted

We hope to hear from you within two weeks; please let us know if the process may take longer.

Best regards,

Reinhard Maurer, PhD
Editorial Board Member
Communications Materials
orcid.org/0000-0002-3004-785X

Reviewer 1

In this paper, the authors show that a resistance drift in the amorphous phase (towards higher resistance) can result from Peierls distortion. With a focus on single-element Antimony-based PCM, they establish such a link with the following observations from computational results: (1) Peierls distortion (split into regular long- and short- bonds in the lattice) occurs during the quenching process, and (2) there is a correlation between lower Peierls distortion and higher current density. The slow increase in distortion in the amorphous material over time can thus slowly shut down these current pathways, thus leading to an increase in resistance over time.

The results presented seem interesting, but the manuscript could be further improved:

We thank the reviewer for the positive assessment of our work. We hope to remedy the mentioned issues with the changes described below.

General points

1. The discussion focuses on the purely amorphous phase of the material. However, resistance drift in PCM is mainly an issue in Multi-Level-Cells (MLCs) which try to achieve several intermediate states. Such intermediate states may have crystalline and amorphous domains, and the current would probably be carried dominantly by electronic states corresponding to the crystalline regions. If this is the case, is the contribution of atomic distortion processes within the purely amorphous regions still a very relevant effect for the resistance drift observed in practice? The authors specifically compare their findings to the magnitude of resistance drift observed in PCMs (page 10, paragraph 2), so a discussion on this would be relevant.

The reviewer is fully correct that transport through crystalline inclusions in the amorphous material is indeed an important alternative to other conduction mechanism like Poole-Frenkel conduction (see e.g. Nardone et al., Journal of Applied Physics 2012). However, even with the presence of crystalline domains within the amorphous region, the total device resistance is not necessarily determined by properties of the highly conductive crystalline regions. Instead, the resistance might still be dominated by the amorphous part. This is because if the conduction occurs via a series of amorphous (high resistance) and crystalline (low resistance) volumes, the higher resistance will determine the total device resistance. Accordingly, different resistance levels in multi-level cells are commonly modeled by changing the size of an amorphous dome (see e.g. Sebastian et al, Nature Communications 2014). Changes in its larger resistance should thus contribute much more to resistance drift than effects related to small crystalline domains in the amorphous matrix.

We have added a comment on this issue to the second paragraph of the discussion section, which reads

[...] Note again that other mechanisms could also contribute to this increase in resistance. Besides effects related to “wrong” homopolar bonds (Raty et al., Nature Communications 2015; Gabardi et al., Physical Review B 2015) or defective

coordinations of germanium atoms (Konstantinou et al., Nature Communications 2019), this could in principle also include effects due to the presence of crystalline inclusions in the amorphous material, which have been proposed to be related to a possible conduction mechanism in the past (Nardone et al., Journal of Applied Physics 2012). However, even in the presence of crystalline regions, the total device resistance through a series of amorphous and crystalline regions should still be heavily dominated by the amorphous part. This is because of the orders of magnitude higher resistivity of these regions. Any changes of the conduction within the purely amorphous regions should thus be much more important for resistance drift than any effect related to crystalline inclusions.

2. The writing is very wordy. For example, too many sentences are used to justify an ab initio approach to transport. The explanation of temperature effects on the bandgap can also be shortened.

As proposed by the reviewer, we have shortened the justification of our ab-initio approach by removing the following sentences or parts of sentences:

1. *At latest in miniaturized devices smaller than 10 nm (Chen et al., 2006), we can hardly imagine a situation where this kind of model describes physical reality. ~~Still, phase change devices with these small dimensions have existed for a rather long time, also showing resistance contrast between the crystalline and amorphous phase. In addition, with the advent of pure antimony as a phase change material (...), the above-mentioned pictures of electronic traps in the amorphous phase based on defective coordinations or homopolar bonds around germanium atoms cannot hold for all PCMs.~~*
2. *[...] without resorting to (semi-)empirical models for the connection between electronic structure and transport. These models are usually based on Poole or Poole-Frenkel effect, field-induced delocalization of tail states, or hopping conduction. In contrast to this approach, we do not a priori assume certain properties of localized states around the Fermi level, but use a non-equilibrium framework to calculate transmission through the device self-consistently. (changed to “[...] without resorting to the (semi-)empirical models for the connection between electronic structure and transport discussed in the introduction section.”)*

We have further shortened the explanation of temperature effects on the band gap as follows: “Furthermore, a reduction of the band gap size due to temperature is often ignored in electronic structure calculations of other PCMs, such that an actual gap might not even exist over the full range of operating temperatures of PCMs, ~~at least not close to the melting point~~ (Cobelli et al., Physical Review Materials 2021). ~~The 0 K results thus do not necessarily reflect experimental reality.~~ We believe that simulations of electronic transport in other PCMs at elevated temperatures are needed to disentangle the contributions of the described variety of mechanisms to resistance drift.”

Comments

3. The authors state that "results obtained using GeTe might not be transferable to PCMs dominated by antimony", but also that "This gives direct evidence that the increase in Peierls-like distortions upon structural relaxation of the PCM glass shown in [10] is indeed a source of resistance drift.", where Ref [10] uses GeTe. Maybe it would be better to cite experimental studies which show some evidence of Peierls distortion in Antimony. For example, the following paper:

<https://onlinelibrary.wiley.com/doi/full/10.1002/adv.202301043>

We thank the reviewer for the good suggestion. Our first statement on GeTe was misleading. We have changed this sentence to “However, the role of atomic structure in general and Peierls-like distortions in particular is not discussed, and it is not obvious how results from GeTe can be transferred to PCMs dominated by antimony.” We have further added a reference to the suggested publication to the discussion of the transferability of our results to other PCMs. It now reads “This gives direct evidence that an increase in Peierls-like distortions indeed increases device resistance. An increase in distortions has been observed upon structural relaxation of PCM glasses in a previous work (Raty et al., Nature Communications 2015), which links our results to resistance drift. For pure antimony, the suppression of resistance drift in 4 nm films has recently been connected to less distorted octahedral-like environments due to Sb/SiO₂ interfaces (Chen et al., Advanced Science 2023).”

4. Figure 1 needs some significant improvements. The figures and font are all too small, part (b) needs a y-axis label (it seems to be the distance, but an axis should still have a label). There must be a way to show the information in part (c) in a clearer way.

Following the reviewer’s recommendations, we have increased the overall font size from 7 to 8. The Nature recommendation is 5 – 7 pt. We have added the y-axis ticks and label in part (b). We have replaced the color map in subfigure (c) (and in the corresponding inset of Fig. 3a), which should show the distribution of Peierls-like distortions in a clearer way.

Fig. 1: Heterogeneous distribution of Peierls-like distortions in amorphous antimony. A more detailed figure caption can be found in the main text.

5. The distortions exist in clusters, rather than being homogeneously distributed, but the authors state that it might be influenced by the small cell size. How large is this influence? Can these clusters be a complete artefact of the finite simulation box?

The reviewer raises a very important question here. Unfortunately, the influence of finite size effects on the size of clusters with different degrees of Peierls-like distortions is very difficult to quantify based only on ab-initio molecular dynamics simulations. Because they are so computationally demanding, these simulations are usually limited to approximately 1000 atoms or fewer. A relevant (factor of two or three) increase in the size of the simulation cell is therefore impossible to simulate with ab-initio simulations in a reasonable amount of time. In Fig. C1, we show preliminary results from an ongoing project, which have been obtained using a machine-learned potential that was fitted to the data from (Holle et al., Physical Review Letters 2024 – accepted). In two systems of supercooled-liquid antimony with 8000 and 64000 atoms (both at 500 K), we again observe the formation of clusters and an inhomogeneous distribution of Peierls-like distortions. The clusters are therefore not simply an artifact of the rather small cell size in the ab-initio molecular dynamics simulations. We further report the correlation function of Peierls-like distortions in Fig. C2, again at a temperature of 500 K, for these two larger system sizes. We find correlations that extend over up to 2 nm at 500 K. Note that compared to the ab-initio MD simulation shown in our manuscript, this length does not increase further when increasing the system size from 8000 to 64000 atoms. Our results show that the ab-initio MD simulation of a rather small cell can reproduce qualitatively the inhomogeneous distribution of Peierls-like distortions, but the longer-range tail in the correlation function that extends up to 2 nm is suppressed in smaller systems due to finite size effects.

Fig. C1: Distribution of Peierls-like distortion in supercooled-liquid antimony at 500 K for two different system sizes (left: 8000 atoms, right: 64000 atoms). Yellow and blue colors indicate less and more distorted regions, respectively.

Fig. C2: r_2/r_1 correlation function as introduced in our manuscript for three different system sizes.

6. Could the mass density oscillation effect be due to internal pressure during the MD simulations, from fixing the positions of the outer contact atoms (which implicitly imposes a fixed cell)? Does it remain if a full geometry-relaxation is performed?

The reviewer's suspicion that the density oscillations could be due to internal pressure is very plausible. However, the density profile is very independent of temperature (see Fig. S2 repeated below), which already indicates that pressure does not have a strong effect.

Fig. S3: Formation of a layered structure in the amorphous thin film. Already at high temperatures above 2000 K, a dense wetting layer forms on top of the (100) tungsten surface that follows the BCC structure of tungsten. Spatial oscillations of density due to the confinement are also visible at high temperatures and become more evident at the melting point of antimony at approximately 900 K and below. The density at the centre of the device ($z = 0$ in the lower sub-figure) increases at lower temperatures, which could be an indication of negative thermal expansion. This effect is also known for other phase change materials.

Fig. S4: Peak positions in the mass density in Sb/W device structures in dependence of electrode distance. The electrode distance was reduced by the percentages encoded in colour, and additional melt-quench MD simulations have been performed. We observe both changes in the peak positions and changes in peak heights upon compression, but no signs of a disappearance of the oscillations in mass density if the system is given enough volume.

To clarify this point further, we performed *ab-initio* MD simulations with different electrode distances, i.e. different amounts of volume given to the PCM. The resulting local density profiles are shown in Fig. S3.

We observe that giving the PCM more or less volume shifts the peak positions of the density oscillations, but the oscillations do not disappear upon giving the amorphous PCM more volume. Finally, we also calculated the stress in our Sb/W device structures in dependence of electrode distance, which is shown in Fig. S4. Overall, the stress is rather small for all electrode densities studied, and reduces to almost zero for the structure studied in our manuscript. Consequentially, a geometry relaxation of the Sb/W structure from our manuscript quenched to 100 K shows only small changes of the cell vectors, and the oscillations in mass density also persist after the relaxation (see Fig. S5 and S6).

We have also added the data discussed here to the supplemental information (local density profiles and stress in dependence of electrode distance, changes in potential energy and cell dimensions during and local density profile after relaxation).

We conclude that the oscillations in mass density are in fact due to the contact to the electrode material, and are not related to pressure-induced effects.

Fig. S5: Diagonal elements of the stress tensor for our Sb/W device structures at 400 K in dependence of electrode distance.

Fig. S6: We find only small changes both in potential energy and in the cell parameters upon relaxation. The structure was relaxed using a BFGS optimizer until all forces were below a threshold of 15 meV/\AA . To this end, DFT calculations were again performed using CP2k with the same settings as for the MD simulations shown in our manuscript. Instead of the more efficient orbital transformation method, we used a standard diagonalization scheme and also included Fermi-Dirac smearing with a width of 300 K.

Fig. S7: The oscillations in mass density persist after relaxation of both the simulation cell and atomic positions. Details of the structural relaxation are given in the caption of Fig. S5.

7. It seems that the potential is not flat in the contacts (Fig. 4c). Maybe this is due to computational limitations on the number of contact atom layers which could be included, but it should be addressed in the text.

The reviewer is correct that we kept the number of layers in the electrode extensions rather low, as more layers would have further increased the already excessive computational cost of our DFT/NEGF calculations. In experiments, one finds a conductivity of tungsten that is approximately two orders of magnitude larger than the conductivity of amorphous antimony in our simulations. This means that the electrostatic difference potential should appear flat in the electrodes in Fig. 4c, which is not the case. The limited number of electrode layers is indeed the most probable reason

for this artifact. The limited number of k-points (Γ -point approximation) could also play a role. We added the following sentence to the caption of Fig. 4 that should clarify this point:

“Note that this potential is not completely flat in the contacts, probably due to the limited number of electrode layers and the limited number of k-points, which are both imposed by the computational demand of our simulations.”

8. "Peierls-like distortions can change the local current by a factor of two (Fig. 4 e), which is of the order of magnitude that is usually observed for resistance drift ... This gives direct evidence that changes in Peierls-like distortions upon structural relaxation of the glass give a relevant contribution to resistance drift. " - I think this is quite a large jump. See the 'general comments' section above.

We thank the reviewer again for this suggestion that further improves the line of argument in our manuscript. We have made several changes to the sentences mentioned above that should close the gap between the first and the second statement. The paragraph now reads:

“This gives direct evidence that an increase of Peierls-like distortions upon structural relaxation of the glass, which has been observed in the past, does in fact increase device resistance. [...] We can therefore link changes in Peierls-like distortions to resistance drift. The factor of two change in local currents mentioned above indicates a sizable contribution of this effect to the overall drift, but further investigation is needed in particular for multicomponent PCMs like GeTe or Ge₂Sb₂Te₅, where different effects have also been proposed.”

9. In Fig. 4b, the current is negative in the electrodes and positive in the PCM. How can that result in a netto positive or negative current? Or is that an artifact of the plotting scheme?

The color code in our original version of Fig. 4b was misleading. Tungsten atoms were indicated with blue diamonds, matching the color of negative currents. We have changed the color map, which now does not include any blue tones.

Minor comments

10. Use `` instead of " for the left open bracket in LaTeX

As kindly proposed by the reviewer, we have changed the quotation marks in LaTeX.

11. Paragraph 3: one mechanisms  one mechanism

12. There are several similar minor typos

11 and 12: We have fixed this and several other typos.

13. "Structurally correlated regions with smaller and larger amounts of distortions evolve at low temperatures." - maybe the authors can rephrase this

We have changed this sentence to “Structurally correlated regions with different degrees of distortion evolve at low temperatures.”

14. Some of the details behind how distortion was measured/defined should be shifted into the main manuscript, since this is important to follow the text

We also thank the reviewer for this recommendation. We have changed the corresponding sentences (first paragraph of the Results section) to “During quenching, we monitored the distances of the six nearest neighbors around each atom. These distances can be further grouped into three long and three short bonds. We also measured the magnitude of Peierls-like distortions using the angular-limited bond length ratio (details in the Methods section). This is the ratio of the average of the three longer distances and the average of the three shorter distances, limited to bonds that have an angle between 155° and 180°.”

15. "We find more and less conductive paths through the glassy phase change material (Fig. 4 b). " - It is not clear what this means, even after looking at Fig. 4b. The authors may mean to say something like "regions of high (red) and low (blue) current density".

We have changed this sentence to “We find regions of high (red) and low (white) current density in the glassy phase change material (Fig. 4b).”

16. Some important information seems to be located in a manuscript the authors currently have under review elsewhere. Maybe phrases like "As shown in Ref [37]" should be rewritten to "as shown using ... [37]" so the results can be more easily understood in isolation

We thank the reviewer for this observation. We have changed two sentences in the following way:

“In addition, we have shown in another previous work on the atomic structure and electronic and optical properties of antimony in dependence of density (Holle et al., Physical Review Letters 2024 – accepted) that the lowest density of 5.9 g/cm³ studied there offers the largest amount of Peierls-like distortions, which promises to yield the largest effect on the electronic structure.”

“As shown in a previous work (Holle et al., Physical Review Letters 2024 – accepted) using density-dependent ab-initio molecular dynamics simulations of antimony, the effect of further lowering the temperature is mainly a slight additional increase in Peierls-like distortions, as the thermal energy is not sufficient for larger reconstructions.”

17. "we melt-quench another model of antimony comprising approximately 1100 atoms that includes tungsten electrodes" - It should be clarified here that the outermost tungsten positions were fixed.

We have added the following sentence to this paragraph for clarification: “The two outermost tungsten layers were fixed in the molecular dynamics simulations in order to resemble the bulk structure of tungsten.”

18. "We use a DFT-based method (details in the Methods section) to calculate the current density" - maybe this should be rephrased, it sounds like DFT is solving the non-equilibrium problem.

We have changed this sentence to “We used an NEGF/DFT method (details in the Methods section) to calculate the current density through our phase change device structure.”

19. "but also exists at the interface of crystalline antimony and tungsten (100). " - Citation missing (this should probably be [3] from the Supplementary Materials)

Again, we thank the reviewer for the careful observation. In fact, the larger distance between the wetting layer and the remainder of the antimony atoms is shown in Fig. S8. Consequentially, we have added a reference to this figure to the mentioned sentence.

Finally, we would like to thank reviewer 1 again for the positive assessment of our work, and the detailed, critical, and very constructive feedback. We think that all the points and changes discussed above have significantly improved our manuscript.

Reviewer 2

In this manuscript, Holle *et al.* performed ab initio simulations of non-equilibrium transport in a model of amorphous antimony (Sb), by combining density functional theory (DFT) with non-equilibrium Green's function (NEGF) calculations. It is widely accepted that electronic transport in the amorphous state of phase-change memory materials is not very well understood, and in-depth investigations, at the atomic level, of all the relevant aspects that govern the transport in such memory devices are missing from the literature. In addition, transport can be correlated to the time-dependent resistance drift phenomenon observed in the amorphous phase of these materials.

Here, the authors are trying to draw connections between atomistic models of amorphous phase-change materials and macroscopic electronic transport characteristics observed in the devices. They show that small changes in the local atomic structure, through Peierls-like distortions, can change rather significantly the local current density, which, in turn, can be correlated with the resistance drift typically observed in the device. Hence, they conclude that such Peierls-like distortions should play a role into the structural relaxation of the amorphous state (Sb in this case as the amorphous structure). In addition, the authors showcase the importance of studying the effects of nano-confinement in phase-change memory devices, and how this can affect all the properties of the material, while also they discuss the relevance of interfaces inside the devices which are very small in size.

The research is very interesting, while the authors have extensive experience in studying phase-change memory materials with experimental and modelling techniques, providing useful contributions to the relevant community in understanding the processes inside phase-change memory devices. The authors provide technical details about the atomistic simulations carried out here, the electronic structure properties of amorphous Sb, the NEGF calculations, and the results about the electronic transport. Also, I appreciate the significant effort in the Discussion section of the manuscript to provide a comprehensive picture about the limitations of this study, the comparisons with different compositions (binary, ternary, etc.), the connections between the results and experimentally-based observations, as well as the suggestions for future work related to the concepts presented here.

We also thank reviewer 2 for the very positive feedback and detailed evaluation of our manuscript. We again hope to remedy the concerns and suggestions with the responses given below.

Below, I express some comments, concerns and suggestions that I would like to ask the authors to consider:

1. The authors mentioned in the last paragraph of the Introduction that “the only existing work that studies non-equilibrium transport in a phase-change material is by Liu & Anantram [36].” a statement related to previous atomistic simulations available in the literature. Apologies if I am mistaken, but I think this is not probably 100% true, since this is not the only existing study. Previously, electronic transport in amorphous and crystalline $\text{Ge}_2\text{Sb}_2\text{Te}_5$ has been studied with ab initio molecular-dynamics simulations and NEGF calculations [AIP Advances 9, 055120 (2019)]. In addition, ab initio simulations have been combined with NEGF calculations to study electromigration processes in liquid GeTe and Sb_2Te_3 [J.

Phys. Chem. C 124, 9599-9603 (2020)]. Again, these studies seem relevant to me, in case I am not wrong.

We thank the reviewer for pointing us at these two very relevant references. Both works should certainly not go unmentioned in our manuscript. We have therefore modified the section of the main text mentioned by the reviewer in the following way: “So far, only very few works exist that study non-equilibrium transport in a phase change material atomistically. The oldest work is by Liu et al. (Liu et al., IEEE Electron Device Letters 2014). The field-dependence of transport in both the metastable rocksalt and amorphous phase of $\text{Ge}_2\text{Sb}_2\text{Te}_5$ has been investigated by Roohforouz et al. (Roohforouz et al., AIP Advances 2019). Electromigration in GeTe and Sb_2Te_3 has been studied using NEGF simulations by Cobelli et al. (Cobelli, Journal of Physical Chemistry C 2020). However, the role of atomic structure in general and Peierls-like distortions in particular is not discussed in these works, and it is not obvious how results from GeTe , $\text{Ge}_2\text{Sb}_2\text{Te}_5$ or Sb_2Te_3 can be transferred to PCMs dominated by antimony.”

2. The authors highlight in the Introduction the necessity for “models and methods that are not based on electronic traps of questionable existence”, while they also say that in miniaturized devices the concept of trapping cannot really describe physical reality. I can imagine that the miniaturization from a device engineering point of view might exclude such processes. But then, the authors in the results section of the manuscript discuss “localized states” in their amorphous Sb model. If here one refers to the spatial electron localization of an electronic state (such as those that appear in a glass near the bottom of the conduction band or the top of the valence band), then a “localized state” can potentially serve as a charge-trapping centre inside the glass structure. I think it would be interesting for the reviewers and the (future) readers if the authors could kindly clarify this.

We thank the reviewer for this comment, which shows us that the mentioning of localized states through Peierls-like distortions in the Results section needs further clarification. The reviewer is correct that in radically miniaturized devices, effects related to electronic traps cannot describe physical reality, as discussed in detail in the introduction section. Yet, such interpretations have been successfully used to describe experimental measurements of large samples. As our simulations are limited to small systems, the results section contains what is rather a Gedankenexperiment: Which effects could our results imply in larger structures? If the structure becomes much larger than the correlation length of Peierls-like distortions (probably around 2 nm at 500 K, see question 5), less distorted regions could, in principle, act as localization and thus charge trapping centers. As we did not yet study the electronic structure of much larger systems, this idea certainly needs further investigation.

We think that the distinction between the situation in radically miniaturized devices on the one hand, and the possibility that localization could occur in less Peierls-like distorted regions in larger structures (tens of nanometers and more) on the other needs more clarification in our manuscript. We therefore added the following sentence to the discussion section: “[While this extent is already rather large for ab-initio molecular dynamics simulations, it is still small compared to phase change cells studied in experiments with sizes often exceeding tens of nanometers.] In contrast to the situation in radically

miniaturized devices described in the introduction, where conduction based on electronic traps is questionable, localized states can well determine transport characteristics in larger structures. [Furthermore, we have to expect more localized states around the Fermi level for PCMs with an actual band gap like $\text{Ge}_2\text{Sb}_2\text{Te}_5$ or GeTe]

3. In Figs. 1(a) and 1(b) the authors show the atomic distances of three short and three longer bonds. For the longer bonds they state a range of values, but the shorter bonds they seem to be almost identical. My question is if this is meaningful. Why do you need to define “three different” short bonds that are almost the same? In other words, what is the statistical significance of these three different short bonds?

The reviewer is fully correct that the bond lengths of the three shorter bonds are indeed rather similar, as indicated by the large overlap of their respective distributions in Fig. 1a. In Fig. 1, nearest neighbors are ordered by distance to the central atom in order to (a) show the continuous emergence of Peierls-like distortions upon melt-quenching, and (b) introduce the definition of the quantity used to measure Peierls-like distortions throughout the rest of the manuscript, i.e. the ratio of long (neighbors 4-6) to short (neighbors 1-3) bond lengths. We would like to stress that here we averaged over the distances to neighbors 4-6 and neighbors 1-3, respectively. This means that in our measure for Peierls-like distortions, we do not further distinguish the three short bonds.

4. In Fig. 1(e), the specific choice of temperature values of 942 K and 2502 K is rather striking. It might sound quite particular, but why not 940 and 2500 K?

We have rounded the values to 950 K and 2500 K, respectively, in line with the other temperature values in the legend.

5. In page 5, the authors bring into the surface the issue of “finite size effects” regarding the extension of the distortions within the periodic cubic simulation cell. They have modeled a glass structure of 728 atoms, which is already quite large for ab initio simulations. Could the authors provide an estimation or a view of what size of glass model should be adequate for such quantifications? Size effects is usually an “easy argument” (still valid), but if one puts it into the context of what is the size needed for the investigations then it becomes more meaningful, especially for comparisons. In other words, a discussion between a qualitative and quantitative comparison.

We also thank reviewer 2 for this very relevant question, which is very similar to question 5 of the first reviewer. As outlined in our answer to that question, we have estimated finite size effects using preliminary results from MD simulations of much larger systems using a machine-learned potential. These results indicate that while the strong correlation of Peierls-like distortions on length scales up to 7 Å is well reproduced in the smaller cell of our ab-initio MD simulation, and extends up to 1 nm, correlations on longer length scales (up to 2 nm in the simulations using a machine-learned potential) are suppressed due to finite size effects. Cell sizes of approximately 4-5 nm should therefore be regarded as a lower limit to accurately model the extension of regions with different degrees of Peierls-like distortions.

6. The choice of the quench rate for the generation of the amorphous Sb model seems a bit arbitrary (9.5 K/ps). Could the authors provide a brief justification regarding this?

We thank the reviewer for this observation. We have added the following sentence to our manuscript: “The quenching rate of 9.5 K/ps used here is the slowest rate that did not lead to an onset of crystallization during quenching in the study by Salinga et al. (Salinga et al., Nature Materials 2018)”.

7. The choice of colour for the text and dash lines in Fig.2 is completely inadequate. It is very difficult to read that text and distinguish the lines in the figure with using this silver-like colour. I strongly recommend to change this, since the readers cannot see those (especially the text written).

We have adjusted the colors in Fig. 2. We hope that the black text is now more readable.

Fig. 2: Heterogeneous distribution of Peierls-like distortions is an equilibrium property and is observed in the supercooled-liquid. The potential energy for the model with a mass density of 5.9 g/cm³ is shown in dependence of temperature during melt-quenching in ab-initio MD simulations with a quenching rate of 9.5 K/ps (dark line). At several different temperatures, branches with constant temperature are created. Here, color indicates the time starting at departure from the melt-quenching trajectory. While full equilibration is observed directly after quenching at 600 K and 700 K, the system begins to deviate from the supercooled-liquid line at approximately 500 K. A full explanation can be found in the main text.

8. In the analysis presented in Fig. 2, the authors extracted specific snapshots of the MD trajectory at temperatures during the quench of the simulated structure, and then run longer MD simulations to equilibrate the model at these temperatures. I am wondering if the authors considered to perform NPT (constant temperature, constant pressure) MD simulations at this stage. The quench simulation is usually adding a residual stress to the simulation box. If one performs an NPT simulation initially for the extracted geometry at the specific temperature, it should help the structure to adjust at the “simulation volume” that is happy to find, and then it could equilibrate smoother. In that way, the structure should find its way to the respective equilibrium volume as well. It would be interesting to hear what is the view of the authors, and how this could affect their structures (especially the Peierls-like distortions) and results? To my mind, this should be relevant regarding such distortions.

The reviewer raises a very important issue here. Unfortunately, the 2nd generation Car-Parrinello method employed for our ab-initio molecular dynamics simulations does not allow for NPT simulations, as the method is slightly dissipative and requires using Langevin dynamics. However, we have very recently studied the effect of a quantity which is very closely related to pressure, the mass density, with a particular focus on Peierls-like distortions (Holle et al., Physical Review Letters 2024 – accepted). The two figures below (Fig. C3 and Fig. C4) show the temperature dependence of Peierls-like distortions in

Fig. C3: Splitting of nearest-neighbour distances into long and short bonds vanishes with increasing density. (a-c) Average distance of the six nearest neighbours in dependence of temperature for six different densities. (d) Angular-limited bond length ratio (ALBLR, as defined in the appendix) in dependence of temperature and density. The ALBLR describes the average ratio of one long to one short bond for each atom. The inset shows the similar and more common angular-limited three body correlation (300 K, 6.07 g/cm³, see appendix).

dependence of density during melt-quenching, as well as the amount of stress in the structures. We observe a gradual increase of Peierls-like distortions at all densities studied, where the onset of distortions is shifted to lower temperatures with increasing density. In general, the stress decreases upon decreasing temperature, but shows an increase below the onset temperature of Peierls-like distortions. Imposing stress on the structure therefore allows tuning the amount of Peierls-like distortions, up to a full removal of this structural feature at a mass density of approximately 6.8 g/cm^3 . As such, the qualitative link between the local amount of Peierls-like distortions should be independent of pressure and should not differ between structures generated using NVT and NPT simulations, but the fraction of regions with different amounts of Peierls-like distortions does depend on pressure.

Fig. C4: Changes in structure are closely accompanied by changes in stress in the system. Depiction of the dependence of the virial stress on temperature and density. The stress is obtained from the CP2k calculations used to obtain the DOS data in Fig. C5. While higher densities generally lead to an increase in stress, we observe an additional increase of stress upon melt-quenching below 800 K at low densities that is absent at the highest densities.

9. The authors throughout the manuscript they make a rather big statement/argument that in contrast to traditional DFT calculations performed at 0K, they have performed simulations at temperatures that are well in range with the experimentally (device operation level) temperatures. I can see the point and I am happy to follow it, especially for higher temperatures, close to the liquid or super-cooled liquid regimes. But then, the authors use this argument for their analysis of a structure at 150 K. I would be a little bit sceptical to believe that a model structure of a glass at 150 K would look significantly different (geometrically) from the same structure quenched to 0 K, for example. MD simulations at these temperatures typically they do not yield at different atomic structures (essentially atoms do not have much chance to move). Unless for amorphous Sb the differences in the

atomic structure for temperatures between 0 – 300 K are that significant. I think the authors need to consolidate this argument with the atomistic models that have generated.

The reviewer is correct that at temperatures as low as 150 K, we cannot expect any larger atomic movements. However, our results from a previous study (Holle et al., Physical Review Letters 2024 – accepted) show that even at this very low temperatures, changes in bond lengths are still possible. In Fig. C3, it can be observed that the distance to the three nearest neighbors still decreases down to temperatures as low as 150 K, enhancing Peierls-like distortions. Following this trend, we can expect significant differences between structures at 0 K and at 300 K, both with respect to the atomic and electronic structure, which we will discuss in more detail below. In particular, we observe differences between the electronic structure at 0 K and 300 K with respect to the number of states at the Fermi level.

10. The same concept also applies for the band gap of amorphous Sb. My first question is: does one expect for amorphous Sb to have a band gap (such as GeTe, for example)? Of course, I follow the argument of the authors that in higher temperatures the band gap of the model is “shrinking”, but in lower temperatures (such as 300 K or 150 K) will the system naturally have a band gap? Because if the answer is yes, then the methodology of choice to study the electronic structure and relevant properties is relatively inadequate for making strong arguments. The authors have discussed this already in the manuscript (at least for the IPR calculations) with respect to the DFT GGA approach, while also they performed calculations with a more accurate, meta-GGA functional. This is all good. But, could it be possible a calculation with a hybrid functional (i.e., inclusion of the Hartree-Fock exchange) to give a result of a model of amorphous Sb with having a well-defined band gap?

We agree with the reviewer that the size of the bandgap of amorphous antimony is very relevant for all our results and conclusions. In a previous work, we studied in detail the electronic structure of amorphous antimony in dependence of temperature and density (Holle et al., Physical Review Letters 2024 – accepted). The data were obtained with the TASK meta-GGA functional, and are shown in Fig. C5. We found that the number of states at the Fermi level is reduced both by a decrease in temperature and in density, but there is no bandgap even at the lowest density and temperature studied (5.90 g/cm³, 150 K).

Fig. C5: An increase in density increases the number of states available for conduction. 16 AIMD models with 728 atoms are quenched from the melt with a constant quenching rate of 9.5 K/ps. The density is varied between 5.90 g/cm³ and 6.83 g/cm³, as illustrated in the top panel. Additionally, previously reported experimental and theoretical densities are indicated with solid white lines (experimental crystalline (Wyckoff 1963, Schiferl 1969), experimental liquid (Crawley 1972), crystalline from DFT calculations using the PBE functional, “low density” studied by in (Salinga et al., Nature Materials 2018)). The white shaded area indicates the range of densities reported in literature for the liquid phase at the melting point. (a) Electronic density of states (DOS) of single snapshots of amorphous antimony at 300 K, where colour encodes mass density. (b) Number of states at the Fermi level in dependence of temperature and density. The dashed line in (b) corresponds to the temperature for which the DOS is plotted in (a), while the dashed line in (a) indicates the energy for which the number of states are visualized in (b).

In Fig. C6, we also show a comparison of the density of states of a model of amorphous antimony calculated using the TASK meta-GGA functional, as well as the density of states of the same structure calculated using the hybrid HSE06 functional. We find very good agreement between both calculations around the Fermi level, which is in line with the accurate estimation of band gaps using the TASK functional shown in literature (Aschebrock and Kümmel, *Physical Review Research* 2019).

Fig. C6: Comparison of the electronic density of states of a system of 728 atoms of antimony obtained using the TASK exchange-correlation functional to a calculation using the hybrid HSE06 functional. While we used CP2k and the libxc library for calculations employing the TASK meta-GGA functional, QuantumATK was used for the reference calculation using HSE06. Details of the calculations in CP2k are given in the methods section of the main body. The PseudoDojo basis set of "high" quality and an energy cutoff of 85 Ha were used in QuantumATK.

11. The way that the authors decided to plot the wavefunctions of selected electronic states in the inset of Fig. 3(a) is rather vague. In the caption it is mentioned that "light and dark colours indicate high and low amplitudes, respectively". But, what is exactly high/low in this case? The authors need to provide a range or values. Moreover, wavefunctions (atomic orbitals) of electronic states obtained from DFT calculations are typically drawn as iso-surfaces, a concept that is useful in order to discuss electron localization or delocalized states. To me, it seems that these figures have been generated with a continuous colour palette within a range of values, which makes it difficult to really assess localization within the amorphous network. Also, the choice of blue and the colouring scheme makes everything difficult to compare. The changes among the snapshots seem very fractional (difficult to see), which then makes difficult to grasp how these connections and arguments could hold true. In other words, the character of localized states the authors claim in their analysis seems very weak, i.e., are the electrons spatially localized in a region inside the glass?

We also thank the reviewer for this suggestion, which helps to further improve Fig. 3. As suggested, we have added a colorbar to Fig. 3a that shows how the wavefunction is mapped to a certain color. We have further changed the color mapping, which should show differences between the different states visualized in Fig. 2 more clearly, and have added a colorbar to this figure. The reviewer is correct that typically, wavefunctions are visualized using isosurface plots, and that this is helpful to assess where exactly a specific state is localized in an amorphous network. For our manuscript, we explicitly decided against this kind of visualization, as we were a) looking for a visualization that can be directly compared to our visualizations of the local amount of Peierls-like distortions, and b) isosurface plots are easily mistaken to show strong localization, even if the state is rather delocalized. Fig. C7 shows isosurface plots of the four states also shown in the main text (Fig. 3).

Fig. C7: Isosurface plots of the four exemplary states shown in Fig. 3 of the main text, again in decreasing order of the IPR (left: most localized, right: least localized). The isovalue was set to $0.014 \text{ \AA}^{-3/2}$ for all four states. Red and green colors indicate a phase of 0 and π , respectively.

We would like to point out again that besides these visualizations which allow only rather qualitative statements about the localization of state, we have also provided quantitative results in our manuscript, where we show the electronic inverse participation ratio (IPR) in Fig. 3a, which e.g. allows concluding that the state shown on the very left in Fig. C7 is in fact more localized than the second-to-left and second-to-right state in the same figure.

12. There is a typo in page 8. The text reads: “The local density of states (Fig. 4) exhibits a layered structure”. I think it should be Fig. 4c.

We have fixed the mentioned figure reference.

13. Throughout the manuscript the authors chose to show their atomistic models with colouring the atoms using a specific palette to describe properties. It would be interesting to see a picture of the atomic configuration as well of the generated models. How do the Sb local environments look like in their model? How do these Peierls-distortions look like at the atomic level?

Before showing the configurations and local structures in more detail, we would like to stress again that our study does not focus on particular single structural elements in the amorphous structure. Our approach is rather statistical, and shows that regions with different amounts of Peierls-like distortions evolve in the supercooled liquid. The most unambiguous signature of this clustering of atoms with different amounts of distortion is the correlation function shown in Fig. 1e.

In Fig. C8, we show again the exemplary snapshot at 150 K from Fig. 1c. Here, lines indicate bonds between atoms, and we have highlighted one exemplary chain of alternating short and long bonds in the system, illustrating Peierls-like distortions on an atomic level. Note that these chains extend over rather long length scales, and can also be found in directions e.g. perpendicular to the one shown in Fig. C8.

Fig. S1: Exemplary configuration of amorphous antimony at 150 K, with a selected chain of atoms illustrating Peierls-like distortions in the amorphous phase. Throughout the structure, we find chains of alternating short and long bonds.

14. My final question is about the transferability regarding the validity of the results by using different DFT packages. The authors presented in the manuscript and supplementary material calculations performed with three different codes (CP2K, Quantum ATK, Quantum Espresso). All these codes can do DFT calculations. CP2K utilizes a mixed Gaussian and plane-waves method with pseudopotentials. Quantum Espresso is strictly plane waves with pseudopotentials. Quantum ATK combines pseudopotentials with LCAO and plane-wave basis sets. A question arises how the different calculations with the different codes compare to each other? The pseudopotentials at each code should be different, while the philosophy of the DFT calculations is also different. Please do not get me wrong, I do not think that the

calculations are problematic, but one needs to be aware of this, especially from the point of view that these are numerical calculations.

The reviewer is correct that the DFT codes used to obtain the results shown in our manuscript (CP2k and QuantumATK) and supplement (Quantum Espresso) are based on different localized and plane-wave basis sets. Furthermore, CP2k uses a mixed Gaussian and plane-waves method, and the question of how the results obtained with these codes compare is fully valid. In Fig. C9, we show a comparison of the density of states of glassy antimony with a density of 5.9 g/cm³ at 150 K, calculated with Quantum Espresso, CP2k, and finally the LCAO-DFT method implemented in QuantumATK. We used a 3×3×3 grid of k-points and the PBE functional in all calculations. We kept the same localized basis sets and cutoff energies that we have already used throughout our manuscript (TZVP basis, pseudopotential of GTH type, and 300 Ry cutoff for CP2k, FHI pseudopotential, DZDP basis, and 150 Ha cutoff for QuantumATK). The plane-wave calculations with Quantum Espresso have been performed using the projector augmented wave method, a Kresse-Joubert ultrasoft pseudopotential from the pslibrary, and 30 Ry and 150 Ry cutoffs for the wavefunction and charge density, respectively. Finally, we would like to stress again that all calculations of electronic structures and transport shown in our manuscript have been performed using the LCAO DFT method implemented in QuantumATK, and not different DFT codes which could, in principle, lead to inconsistencies. We chose CP2k for our ab-initio MD simulations due to the superior efficiency of the 2nd generation Car-Parrinello MD method compared to Born-Oppenheimer MD. Quantum Espresso was used for calculations of the crystalline tungsten/antimony interface and corresponding bonding analyses, as the LOBSTER code interfaces only with plane-wave DFT codes like Quantum Espresso or VASP.

Fig C9: Comparison of the density of states of amorphous antimony at 150 K, calculated with three different DFT codes. Overall, we observe very good agreement between the electronic structures calculated with Quantum Espresso (plane wave DFT), CP2k (mixed Gaussian/plane wave), and QuantumATK (LCAO). Details of the calculations are given above.

The research presented in this manuscript is certainly very topical. The current study contributes to the efforts to rationalize the electronic transport and resistance drift in amorphous phase-change materials. In addition, the authors provide useful insights for future research directions in the field, which is very important for continuation. At the same time, within the current manuscript, it is necessary for some things to be clarified, and provide a more coherent picture.

Overall, I would be happy to potentially recommend the publication of this manuscript in Communications Materials, after the authors have tried to respond to my comments and suggestions.

We would also like to thank reviewer 2 for the extensive and helpful feedback, which has enabled us to further improve our manuscript. We are looking forward to the recommendation for publication in Nature Communications Materials.

Finally, we would like to note that Nature Communications Materials uses a transparent peer review process. This means that all the information contained in our responses to the reviewers' comments will be published together with the manuscript and supplemental information. In particular, this also holds for the preliminary results from a different and ongoing project that we included in our response to question 5 of reviewer 1. As a full description of the machine-learned potential including all validation data would certainly be beyond the scope of this publication, we feel it is appropriate to not include these preliminary results in the SI. Instead, details about the machine-learned potential will be given in a later publication that is dedicated to that method.

Effect of Peierls-like distortions on transport in amorphous phase change devices

Reviewer 1

The authors satisfactorily addressed my comments. I am happy with the revised version of the manuscript.

We would like to thank Reviewer 1 both for the positive feedback and a very constructive review process. We think that we could improve our manuscript substantially with the comments and suggestions.

Reviewer 2

In the rebuttal document, the authors considered carefully all the requests and concerns raised by both the reviewers, providing very detailed answers and explanations. Then, a portion of these answers was inserted in the revised manuscript. A major part of the answers was based on evidence and analyses from another accepted manuscript (in Phys. Rev. Lett.) and a second (yet) unpublished study. I think this is okay, but it is interesting to highlight that for some of the arguments constructed in this manuscript there was a need for extra studies.

Overall, this is a very interesting paper and it corresponds to another piece of the puzzle that the authors (and others) are trying to solve regarding the resistance-drift issue in amorphous phase-change memory devices. Therefore, I would be happy to recommend the publication of the revised manuscript in Communications Materials.

Nevertheless, for the sake of the scientific discussion I have some further comments and observations regarding the responses:

- **Reviewer 2 Q10:** The authors provided further evidence and calculations of higher level of theory to show that their model of amorphous Sb does not have a band gap (essentially a metallic behaviour). I agree and this is all fine, as well as very nicely demonstrated. But one of the questions in the Reviewer report still remains to be answered: "Does one expect for amorphous Sb to have a band gap?", in other words what is the experimentally measured value (if any) of the band gap for the material under study?

We would like to thank Reviewer 2 for again clarifying this very important question. Indeed, arriving at a conclusive answer from literature data whether melt-quenched amorphous antimony has a band gap (or rather, mobility gap) is difficult. This is mainly due to the extremely rapid crystallization of antimony, which hinders experimental investigations of the electronic structure in the melt-quenched amorphous state. On the one hand, the amorphous phase can be stabilized by confinement (Salinga et al., Nature Materials 2018). However, film thicknesses of only a few nanometers are necessary to achieve retention times of the order of seconds or more at room temperature. Of course, this does not leave the electronic structure of the material unaltered, but widens the pseudo-gap and could potentially lead to a mobility gap that is absent in the bulk.

On the other hand, there are several studies on the electrical conductivity and photoemission of as-deposited samples of antimony. Both the deposition and characterization have typically been conducted at low temperatures to stabilize the as-deposited amorphous state. The oldest results go back to Suhrmann and Berndt (Zeitschrift für Physik 1940), who studied antimony films with thicknesses of 35 nm and 250 nm, deposited at approximately 80 K. The authors found an increasing conductivity with increasing temperature, which they interpreted as semiconducting behavior. Photoemission spectra measured by Taft and Apker (Physical Review 1954) indicated a very small band gap in as-deposited amorphous films of the order of 0.1 eV. This is in line with later data by Bringans and Sutton (Solid State Communications 1976), which indicated an optical gap of 0.2 eV at 77 K in as-deposited amorphous Sb, using rather thick 100 nm films. Consequentially, Hauser (Physics Review B 1974) and Mackintosh et al. (Physica B+C 1983) assumed hopping transport to model the temperature dependence of conductivity in (again as-deposited) amorphous films. Unfortunately, all these studies have in common that they used the as-deposited amorphous state, which does not necessarily share the structural or electronic properties of an amorphous state quenched from the melt. However, increasing conductivity with increasing temperature was also observed for the melt-quenched samples in the study by Salinga et al. (Nature Materials 2018) with an apparent activation energy of the order of 0.1 eV, roughly in line with the data for the as-deposited samples.

As a further complication, we would like to note that an increasing conductivity with increasing temperature is not an unambiguous proof of the existence of a mobility gap. In the figure below, we show the temperature dependence of current through the device of our manuscript at a bias of 0.85 eV (also as used in our manuscript). Although there is no indication of a mobility gap in the electronic structure of amorphous antimony in our simulations, we still find an apparent temperature activation. This is because of an increase in the number of states at the Fermi level with increasing temperature (see e.g. Alekseev et al., Soviet Physics Uspekhi 1980). This effect cannot easily be distinguished from the usual temperature-activated transport in semiconductors based solely on the temperature dependence of conductivity.

Apparent temperature activation of transport in our Sb/W device with a bias of 0.85 V.

*Furthermore, it is well known that a negative temperature coefficient of conductivity can occur in strongly disordered metallic systems and amorphous metals. At the beginning of the 1970s, Mooij found a (negative) correlation between this temperature coefficient and the residual resistivity, i.e. the resistivity of a metallic system extrapolated to zero temperature (Mooij, *physica status solidi (a)* 1973). Above a residual resistivity of approximately 150 $\mu\Omega$ cm, the temperature coefficient of conductivity becomes negative. A popular explanation is that of Anderson localization, where random lattice deformations in the limit of strong disorder lead to exponential localization of electronic wave functions. While one of the oldest works in this topic goes back as far as to Kaveh and Mott (*Journal of Physics C: Solid State Physics*, 1982), the origin of Mooij's observation is still actively discussed today (see e.g. Ciuchi et al., *npj Quantum Materials* 2018). In particular, effects of strong disorder have also been related to a negative temperature coefficient of conductivity in a crystalline phase change material (Siegrist et al, *Nature Materials* 2011), with a critical resistivity of 2-3 m Ω cm. Both further experimental and simulation studies on pure amorphous antimony will be needed to disentangle all the different possible sources of a negative temperature coefficient of conductivity mentioned above.*

- **Reviewer 2 Q11:** The authors argued against plotting the wavefunctions obtained from the DFT calculations by utilizing isosurfaces using two reasons in their response. Argument (a) is valid and I get it - the fact that they are trying to compare them consistently with the visualizations of the Peirls-like distortions. However, with argument (b) I beg to differ - if one uses isosurfaces to claim spatial localization for electronic states that are delocalized, then there is something fundamentally wrong in the logic of the user. And as the authors argued in their favour later on, the IPR cannot lie.

We agree with this clarification of Reviewer 2. We didn't mean to discredit using isosurfaces to illustrate the spatial extent of the wavefunction in general. We have further added the isosurface plots from our last response letter to the supplemental information (Fig. S15).

- **Reviewer 2 Q13:** If the authors would like to follow a "statistical" approach, then simulations in many independent models are required to gather statistics. For amorphous materials it is challenging to capture the probability of bond formation and to obtain distributions of their properties via a single numerical study. But, I am sure the authors are in the right path regarding this with their follow-up ML-based study.

Of course, the reviewer is correct that multiple independent MD simulations are necessary to gather statistics. We take this comment as additional encouragement for our ongoing project utilizing ML-based simulations.

- **Reviewer 2 Q14:** The mention of the LOBSTER code seems new to me here. I might have missed it somewhere with respect to the analysis presented in the manuscript.

Actually, we explain the details of the LOBSTER calculations in the supplemental information (caption of Fig. S9) and referenced these calculations in the Results section of the main text (subsection "Influence of Peierls-like distortions on the local current density").

Finally, we would also like to thank Reviewer 2 again for a very constructive discussion and the recommendation for publication in Nature Communications Materials. We truly appreciate the valuable input of both reviewers throughout the review process.